# Structural insights into PA3488-mediated inactivation of *Pseudomonas aeruginosa* PldA

Xiaoyun Yang [1,2,4], Zongqiang Li [1,3,4], Liang Zhao[1,3,4], Zhun She[2], Zengqiang Gao[2], Sen-Fang Sui [1,3] ✉, Yuhui Dong [2] ✉ & Yanhua Li [2] ✉

PldA, a phospholipase D (PLD) effector, catalyzes hydrolysis of the phosphodiester bonds of glycerophospholipids—the main component of cell membranes—and assists the invasion of the opportunistic pathogen *Pseudomonas aeruginosa*. As a cognate immunity protein, PA3488 can inhibit the activity of PldA to avoid self-toxicity. However, the precise inhibitory mechanism remains elusive. We determine the crystal structures of full-length and truncated PldA and the cryogenic electron microscopy structure of the PldA–PA3488 complex. Structural analysis reveals that there are different intermediates of PldA between the "open" and "closed" states of the catalytic pocket, accompanied by significant conformational changes in the "lid" region and the peripheral helical domain. Through structure-based mutational analysis, we identify the key residues responsible for the enzymatic activity of PldA. Together, these data provide an insight into the molecular mechanisms of PldA invasion and its neutralization by PA3488, aiding future design of PLD-targeted inhibitors and drugs.

Phospholipase D (PLD) is a ubiquitous class of enzyme found in organisms ranging from viruses and bacteria to yeasts, plants, and animals[1]. It is capable of hydrolyzing the phosphodiester bonds of glycerophospholipids, resulting in the production of their free head groups and phosphatidic acid (PA)[2]. PA is regarded not only as an important structural element of membranes but also as a secondary messenger of signal transduction[3]. It has been shown that the PA generated by PLD is associated with various physiological and pathological processes, such as cellular signaling, metabolism, inflammation, and tumorigenesis[1,4,5]. Therefore, PLDs in mammalian cells and pathogenic organisms are considered to be valuable therapeutic targets for several human diseases including infectious diseases, neurodegenerative disorders, and cancers[5–8].

To date, many PLDs have been identified in prokaryotic and eukaryotic cells, including in the genus *Streptomyces*[9], *Bacillus cereus*[10], *Escherichia coli*[11], *Pseudomonas aeruginosa*[12], *Arabidopsis thaliana*[13,14], and various mammals[15,16]. In prokaryotic cells, there is commonly one isoform PLD and amino acid sequences of that PLD exhibiting 68.6–85.57% identity within a given genus such as *Streptomyces*. In contrast, low identity, about 7.89–20.63%, is shown between different genera[17]. In mammalian cells, there are six isoforms of PLD, namely canonical PLD1 and PLD2, PLD3, PLD4, PLD5, and PLD6[8]. In plant cells, there is a much larger number of PLD isoforms[18,19].

Despite the poor PLD sequence similarity between different organisms, a conserved sequence motif, HxK(xxxx)$_n$D (in which "x" refers to any amino acid residue, known as the HKD motif), has been

[1]Department of Biology, Southern University of Science and Technology, Shenzhen 518055 Guangdong Province, China. [2]Multidiscipline Research Center, Institute of High Energy Physics, Chinese Academy of Sciences, Beijing, China. [3]State Key Laboratory of Membrane Biology, Beijing Advanced Innovation Center for Structural Biology, Beijing Frontier Research Center for Biological Structure, Tsinghua-Peking Center for Life Sciences, School of Life Sciences, Tsinghua University, Beijing 100084, China. [4]These authors contributed equally: Xiaoyun Yang, Zongqiang Li, Liang Zhao. ✉e-mail: suisf@mail.tsinghua.edu.cn; dongyh@ihep.ac.cn; yhli@ihep.ac.cn

identified among various prokaryotic and eukaryotic PLD isoforms[20]. Most of these proteins contain two HKD motifs, which pack together to form the core of the catalytic domain[21]. The current model of PLD catalysis is a two-step "ping-pong" mechanism. Residues of histidine (H), lysine (K), and aspartic acid (D) are directly involved in hydrolysis of the phosphodiester bonds of phospholipids. The histidine residue from one motif acts as a nucleophile that attacks the phosphorus atom of the substrate, forming a phosphoenzyme intermediate; then, a histidine from the other motif functions as a general acid in the cleavage of the phosphodiester bond[20,22].

In addition to the two HKD motifs located well in the middle of the sequence, PLDs from eukaryotes typically encode an N-terminal phox homology/pleckstrin homology (PX/PH) or C2 domain[23–25]. Based on their N-terminal domains, PLDs can be grouped into PX/PH-PLDs and C2-PLDs. Although the various PLDs differ significantly in their N-terminal domains, these domains are closely related to the binding of membranes and lipids[1,26,27]. In addition, it has been reported that the C2 domain is required for the in vitro activity of C2-PLD, and its activity depends on the concentration of $Ca^{2+}$ [28], whereas the activity of PX/PH-PLDs seems to be independent of its PX/PH domain[29,30].

Prokaryotic PLDs normally have low sequence identity to the eukaryotic PLDs, with one exception: PldA encoded by the PLD-like gene-*pldA* in *P. aeruginosa*. At 122 kDa PldA, is comparatively large for a bacterial PLD[2], and sequence analysis has revealed that it has no PX/PH or C2 domains. In *P. aeruginosa*, PldA has been identified as a T6SS-

dependent antibacterial effector, and it is secreted in vitro to gain advantage over inter-bacterial competition and eukaryotic host infection[2,31]. Previous studies have suggested that PldA promotes the invasion of *P. aeruginosa* into human epithelial cells through the PI3K/Akt pathway[31–33], and it is thus considered to be a key effector for the virulence of *P. aeruginosa* toward human beings. PA3488 is the corresponding immunity protein of PldA, and this can neutralize the virulence of PldA[2]. However, the detailed mechanism of this remains elusive on account of a lack of knowledge of the full-length PldA and PldA–PA3488 structures.

Here, we present the crystal structures of full-length and truncated PldA and the cryogenic electron microscopy (cryo-EM) structure of the PldA–PA3488 complex. These structures show for the first time the open, pre-open, and pre-closed conformations of PLD, which are likely to represent three states of the enzyme's function. These structures and their supporting functional data reveal the architectural features of PldA and unravel in atomic detail how PA3488 binds with and inactivates PldA. These results advance our current understanding of PLDs and provide useful information for the design of drugs targeting PldA.

## Results

### Overall structure of PldA

The crystal structure of full-length PldA (named PldA^FL) was determined at a resolution of 2.1 Å using molecular replacement with *A. thaliana* PLDα (PDB ID: 6KZ9)[24] as a search model (Fig. 1b,

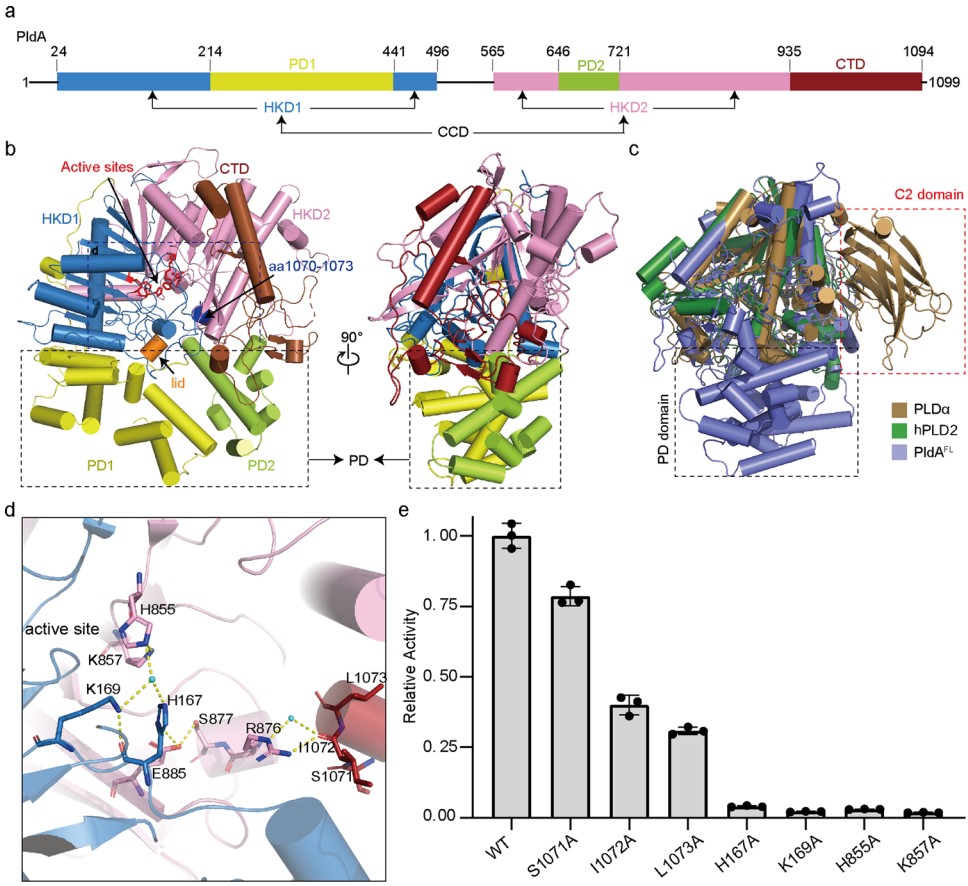

**Fig. 1 | Overall structure of PldAFL. a** Schematic representation of the domain structures of PldA. The PD1, PD2, HKD1, HKD2, CTD, lid, active sites, and aa 1070–1073 are colored in yellow, limon, sky blue, pink, firebrick, orange, red and blue, respectively. **b** Model of PldA with two views. The colors of the domains are the same as those in (**a**). **c** Superposition of the PldA^FL (slate), hPLD2 (PDB: 6OHO, forest), and *A. thaliana* PLDα (PDB: 6KZ9, sand). The distinguishing features of PLDα and PldA are the C2 domain (red dotted line) and PD domain (black dotted line). **d** The C-terminal Ile^1072 forms a hydrogen bond to the Arg^876 from the α24 helix and its vicinity with the active site. The C-terminal and catalytic residues are labeled as sticks, while waters are labeled as spheres. Dashed lines represent hydrogen-bonding interactions. **e** Enzymatic activity assays of key amino acid mutants compared with the wild-type PldA. Results are means ± SD, *n* = 3 biologically independent samples. Dots indicate the corresponding data distribution. Source data are provided as a Source data file.

Supplementary Table 1 and Supplementary Fig. 1a). PldA is mainly composed of 27 α-helices, 25 β-strands, twelve $3_{10}$ helices, and many loops, folding into three regions: the core catalytic domain (CCD, aa 24–935), the peripheral helical domain (PD, aa 215–441 + aa 647–721), and a small C-terminal domain (CTD, aa 936–1094) (Fig. 1a, b). The CCD and PD domains can be further divided into two subdomains-HKD1 (aa 24–496) and HKD2 (aa 565–935), PD1 (aa 215–441) and PD2 (aa 647–721), respectively.

PldA belongs to the family of proteins containing the HKD motif, and it is reminiscent of published eukaryotic and prokaryotic homologs[21,23–25]. Sequence alignment and structural superposition of HKD domains has suggested that PldA is more closely related to the eukaryotic PLDs (Fig. 1c and Supplementary Fig. 2a). In particular, the CCD of PldA is very similar to its human and plant homologs (Fig. 1c and Supplementary Fig. 2a, b, and f–h), and PldA contains a CTD, which is absent from prokaryote PLDs.

Surrounding the CCD, eukaryotic PLDs normally include an auxiliary N-terminal tandem PX–PH or C2 domain[24,25]. In contrast, in PldA, there is an auxiliary PD composed of many interspersed peripheral helices; this is not present in eukaryotic PLDs, and it may introduce unique features (Fig. 1b, c). The B factor (The scores are negatively correlated with the stability of one structure) of the PD region is extremely high, suggesting that this region is structurally dynamic (Supplementary Fig. 2c); in line with this observation, molecular dynamics simulations of PldA$^{FL}$ showed that the positions of these helices fluctuate (Supplementary Fig. 2d, e). These helices lie at the opposite side of the CCD and are lined mostly with hydrophobic amino acids, which may play a role in regulating substrate entry and enzyme activity (Fig. 1b, c).

The CTD of PldA adopts an extended conformation comprising a long α26, a pair of antiparallel β-sheets (β23–β24), several $3_{10}$ helices, and interconnecting loops. The C-terminal residues (SIL$^{1071–1073}$ in PldA) form a short α-helix that protrudes into the cavity pocket (Fig. 1d). Interestingly, the positions and conformations of these residues in the active center are virtually identical to those of human PLDs (hPLDs) and plant PLD (PLDα) (VWT in hPLDs, ILT in PLDα)[23–25], although the differences in their sequences rule out any homology. In PldA, the main chain carbonyl oxygen of Ile$^{1072}$ forms a hydrogen bond to the side chain of Arg$^{876}$ from the α24 helix, which is adjacent to the conserved HKD domain (Fig. 1d). Considering the spatial proximity of this region to the active site, we individually mutated these amino acids to alanine, and the biochemical results demonstrated that the enzymatic activity of the S1071A, I1072A, and L1073A mutants was reduced to ~80%, 40%, and 30% wt activity, respectively (Fig. 1e). This is consistent with previous studies in which mutations in any of these corresponding C-terminal residues partially inactivated hPLDs[24,28,34], suggesting that the C-terminal region is important for the proper functioning of PldA.

## Eukaryotic-like catalytic core of PldA

As noted, as a member of the PLD family, PldA possesses the highly conserved HxK(xxxx)$_n$D consensus sequence. Close examination indicates that the reactive His$^{167}$–His$^{855}$ pair resides in the loop connecting β4–β5 of the HKD1 domain and β17–β18 of the HKD2 domain, spatially in the vicinity of catalytic Lys$^{169}$ and Lys$^{857}$, respectively (Fig. 2a). The side-chain hydroxyl oxygen of Ser$^{870}$ forms a hydrogen bond to the carbonyl oxygen of Glu$^{885}$ from the HKD2 subdomain, in turn hydrogen bonding to the imidazole ring of the conserved His$^{167}$ from the HKD1 subdomain. Symmetrically, an aspartic acid (Asp$^{467}$) from the HKD1 subdomain, instead of a glutamic acid, forms a hydrogen bond to the other histidine (His$^{855}$) from the HKD2 subdomain (Fig. 2a). Moreover, the imidazole group of His$^{855}$ from the HKD2 subdomain, mediated by two water molecules (W1 and W2), indirectly forms a three-hydrogen-bond network with the His$^{167}$–Lys$^{169}$–Asp$^{184}$ pair, thus stabilizing the active site (Fig. 2a). The two HKD subdomains are joined together at the active site mainly by

these extensive hydrogen-bonded interactions. The conformation of the active site region is almost identical to that in eukaryotic PLDs rather than prokaryotic PLDs, indicating a eukaryotic-like catalytically active PLD conformation of PldA (Supplementary Fig. 2a, b). Consistent with its conservation, mutations of the "HK" residues resulted in a complete loss of PldA activity (Fig. 1e). By comparison to the recently published crystal structure of plant PLDα, we also found a species-specific "lid" region composed of residues 117–130 (Supplementary Fig. 3b), which may be used as a switch to regulate the active center. Structural superposition with the "closed" (PDB:6KZ9) and "open" (PDB:6KZ8) A. thaliana PLDα suggested that the lid of PldA$^{FL}$ is far from the active center of PLDα and is closer to the "open" state (Fig. 2b)[24]; hereafter, we refer to PldA$^{FL}$ as being in the "pre-open" state.

## Substrate-binding pocket of PldA

In contrast to the substrate preferences of eukaryotic PLDs, PldA can accommodate a wide range of phospholipids as substrates, with variable hydrolysis rates[35], which is more similar to the hydrolysis features of bacterial PLD (bPLD) from Streptomyces sp. strain PMF[36]. To obtain an insight into the substrate specificity of PLDs, the molecular surfaces and substrate-binding pockets (composed of residues of <5 Å surrounding the conserved His and Lys) of PldA, hPLD2, PLDα, and bPLD were compared. The results suggested that the entrance of the substrate-binding pocket of bPLD is the shallowest and most open, which explains its broad accessibility to phospholipids with headgroups of variable size (Supplementary Fig. 3). In particular, the periphery helices of the PD region in PldA, which surround the entrance of the pocket and distinguish PldA from other PLDs (Fig. 1b, c), have relatively higher structural flexibility (Supplementary Fig. 2c–e), which leads us to speculate that these helices may form the first barrier at the entrance of the substrate; further, they may submit the lipids to the active site in the correct conformation for binding and cleavage.

PldA harbors a large molecular weight and degrades easily (Supplementary Fig. 4), considering the possible poor diffraction of PldA$^{FL}$ crystals, we simultaneously determined the structure of PldA using in situ proteolysis (referred to as PldA$^{truncate}$, the regions containing aa 119–125, aa 221–451 and aa 523–539 were missed in the density due to the flexibility). The structure was determined at 3.05 Å using PldA$^{FL}$ as the search model. In the PldA$^{truncate}$ (Fig. 2a, Supplementary Table 1, and Supplementary Fig. 1b), according to the analysis of the truncated PldA crystal using a time of flight mass spectrometer (MALDI-TOF/TOF UltraflextremeTM, Brucker, Germany, refer to the Source Data for details), the aa 363–411 residues are removed. PldA$^{truncate}$ exhibits an even more open conformation than that found in PldA$^{FL}$ (Fig. 3a). Enzymatic kinetics experiment also showed that the enzymatic efficiency of PldA$^{truncate}$ is about 2.7 times higher than that of PldA$^{FL}$, suggesting the active site is more accessible for substrate in PldA$^{truncate}$ (Supplementary Fig. 5). Close inspection indicates that the periphery helix α11 of PldA$^{truncate}$ rotates ~120° (based on Cα of Leu$^{501}$) relative to the full-length PldA structure and finally stabilized by α18' and α19' from a symmetric PldA' in the unit cell (Supplementary Fig. 6), and simultaneously two PD subdomains separate in two opposite directions, leaving the active site fully exposed (Fig. 3b–d). In PldA$^{FL}$, the α11 helix stacks against the α4 and α9 helices, and these two helices clamp the α3 helix connected to the lid region, thus stabilizing its conformation (Fig. 3c, f). The altered configuration of the α11 helix of truncated PldA eliminates the steric hindrance imposed on the α4 and α9 helices, causing α3 to deviate from its original position. Consequently, the lid region neighboring the α3 helix is also shifted away from the active site (Fig. 3e, f). These movements ultimately lead to a highly open conformation with a solvent-accessible active site (Fig. 3a–f).

We identified a Ca$^{2+}$-binding site coordinated by Asp$^{194}$, Gln$^{928}$, Gly$^{929}$, and Gly$^{931}$ in the PldA$^{truncate}$ structure (Supplementary Fig. 7a, b). The existence of Ca$^{2+}$ in PldA$^{truncate}$ was also confirmed by micro X-ray

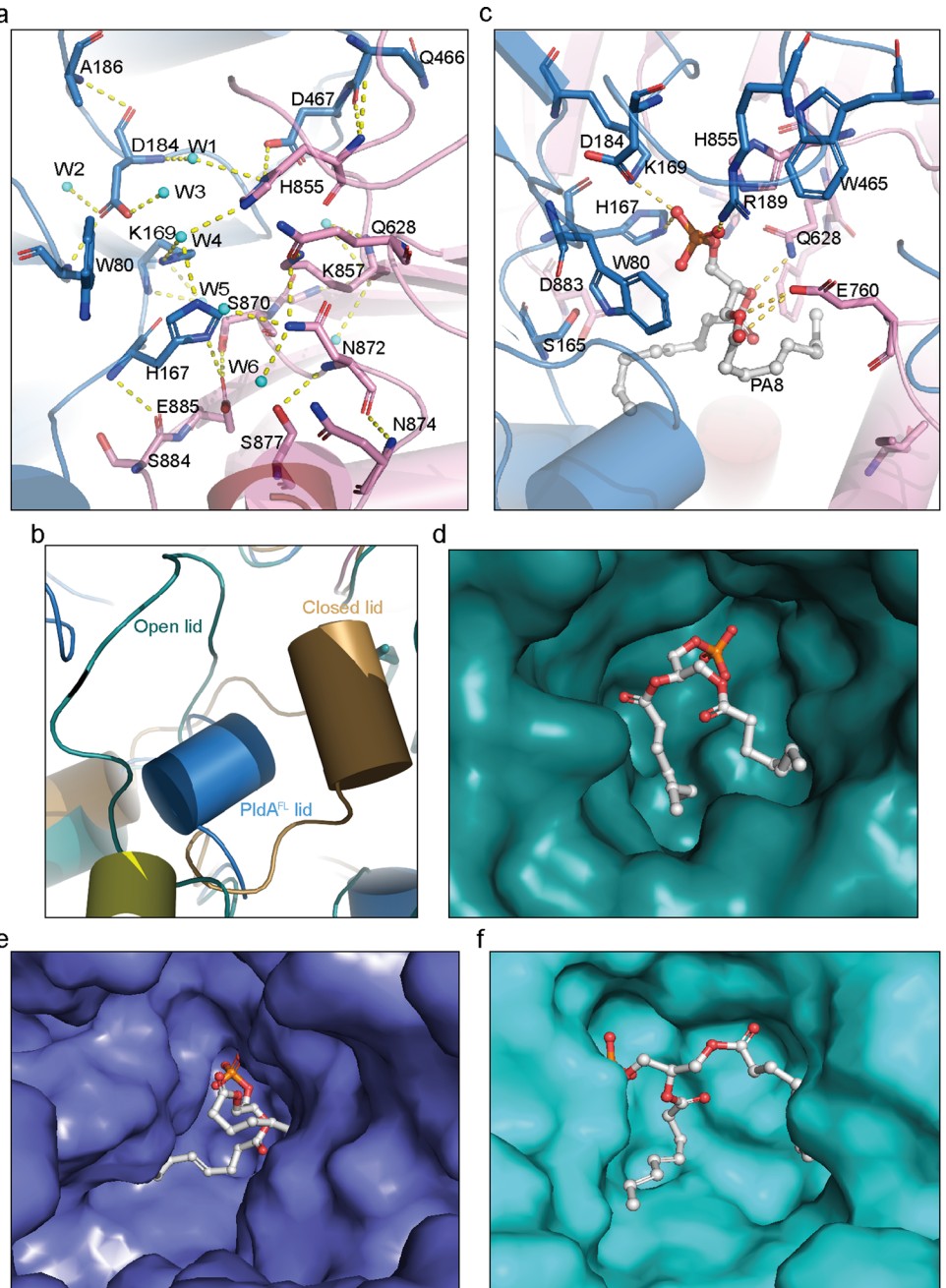

**Fig. 2 | Catalytic pocket of PldA. a** The hydrogen-bonding network present in the active site of PldA. Several water molecules (W1-W6) are involved in the hydrogen bond interaction with active site of PldA. **b** Comparison of the lid regions of PldA^FL (slate) with closed (sand) and open *A. thaliana* PLDα (dark teal). **c–f** The docking result shows the binding of PA8 in the catalytic pocket of PldA^FL (**c**), and surface views showing the binding of PA8 in the catalytic pocket of *A. thaliana* PLDα (6KZ8) (**d**), PldA^FL (**e**), and PldA^truncate (**f**).

fluorescence (μ-XRF) analysis (Supplementary Fig. 7d). Unlike the plant PLD-PLDα, whose activity depends on the concentration of Ca²⁺, the Ca²⁺ concentration in PldA doesn't affect its activity (Supplementary Fig. 7c). This is probably because the Ca²⁺ binding site of PldA^truncate is far away from the catalytic core and may function only in stabling the structure of PldA^truncate (Supplementary Fig. 7a). Besides, the corresponding position in PldA^FL does not contain Ca²⁺, indicating the dispensable role of Ca²⁺ for the activity of PldA.

The predicted model of PldA by AlphaFold2 (https://alphafold.ebi.ac.uk/) is shown in Supplementary Fig. 8a. From the structure, we found that the bulky part of PD1 domain is predicted with low pLDDT (predicted Local Distance Difference Test) scores and high PAE (predicted aligned error) values (Supplementary Fig. 8a, b), suggesting that

this region is highly flexible. The superposition of AlphaFold2 predicted PldA, PldA^FL and PldA^truncate (Supplementary Fig. 8c) shows that the catalytic core of the three structures is well aligned, the PD2 domains are slightly different. However, the PD1 domains and the linker regions between two HKD domains are dramatically different from each other (Supplementary Fig. 8d–f), probably reflecting the dynamic feature of the PD1 domain to regulate PldA's activity.

To examine the specificity of PldA between the different substrates and the binding and orientation of the substrate phospholipid in the active site, we set out to cocrystallize PldA with various substrates. Despite extensive efforts, these experiments have not been successful to date. We therefore used AutoDock to model the reaction product PA8 (di8:0 PA) with PldA^FL and PldA^truncate. The docking results

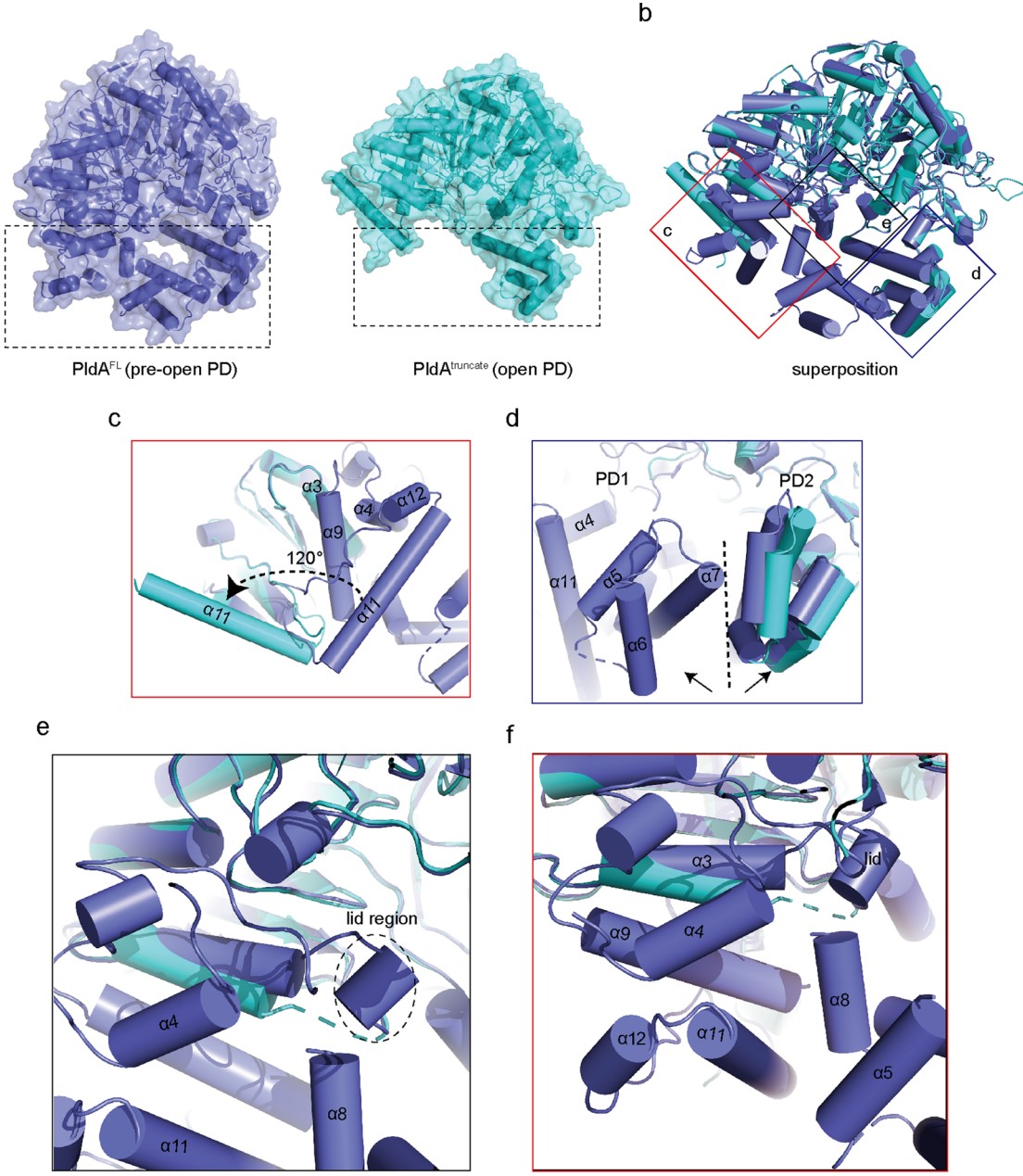

**Fig. 3 | Structural comparison of PldA$^{FL}$ and PldA$^{truncate}$. a** Structural comparison of PldA$^{FL}$ (left panel) and PldA$^{truncate}$ (right panel) shown with a transparent surface. **b**–**f** Superposition of PldA$^{FL}$ (slate) and PldA$^{truncate}$ (cyan) (**b**) indicates the conformational changes of the α11 helix (**c**), PD domain (**d**), lid region (**e**), and the concerted conformational change of the α11 helix and the lid region (**f**).

indicated that PA8 binds to the deeply grooved catalytic pockets, and the PA8 binding mode is highly similar to the PA8–PLDα complex (Fig. 2d–f). The negatively charged phosphate head group of PA8 is located at the bottom of the pocket and is held in this position by extensive hydrogen bonds and salt bridges (Fig. 2c). In detail, the phosphate group forms hydrogen bonds with His$^{167}$ and His$^{855}$; simultaneously, the side chains of the Lys$^{169}$ and Arg$^{189}$ residues neutralize the negative charge on the substrate by forming hydrogen bonds with the phosphate oxygen atoms. Other nearby residues including Trp$^{80}$, Phe$^{629}$, Glu$^{760}$, Ser$^{165}$, Asp$^{184}$, Trp$^{465}$, Asn$^{872}$, and Asp$^{883}$ stabilize the active site and substrate through hydrophilic or hydrophobic interactions (Fig. 2c). In addition, identified with the "open" state of PldA$^{truncate}$, the PA8 molecule exhibits a more relaxed conformation in the catalytic pocket formed in PldA$^{truncate}$ (Fig. 2d–f), implying that it is possible for

PldA to form a more open state when reacting with the substrate; therefore, we refer to PldA$^{truncate}$ to as being in the "open" state.

## Cryo-EM structure of PldA–PA3488

To understand the structural basis of the intrinsic activity of PldA and the way in which PA3488 inactivates PldA, we next determined the cryo-EM structure of the PldA–PA3488 complex by single-particle analysis with a resolution of 3.05 Å (Fig. 4a, b and Supplementary Fig. 9). The structure of the PldA–PA3488 complex indicates that both proteins tightly interact with each other (Fig. 4a, b), burying a ~2020 Å$^2$ solvent-accessible surface area, and PA3488 acts like a crab claw to clamp the HKD2 and CTD domains of PldA. After binding PA3488, the lid composing of aa 117–130 moves toward the active region relative to the PldA$^{FL}$ (Fig. 5a, c), while the two long α helices of PA3488 move

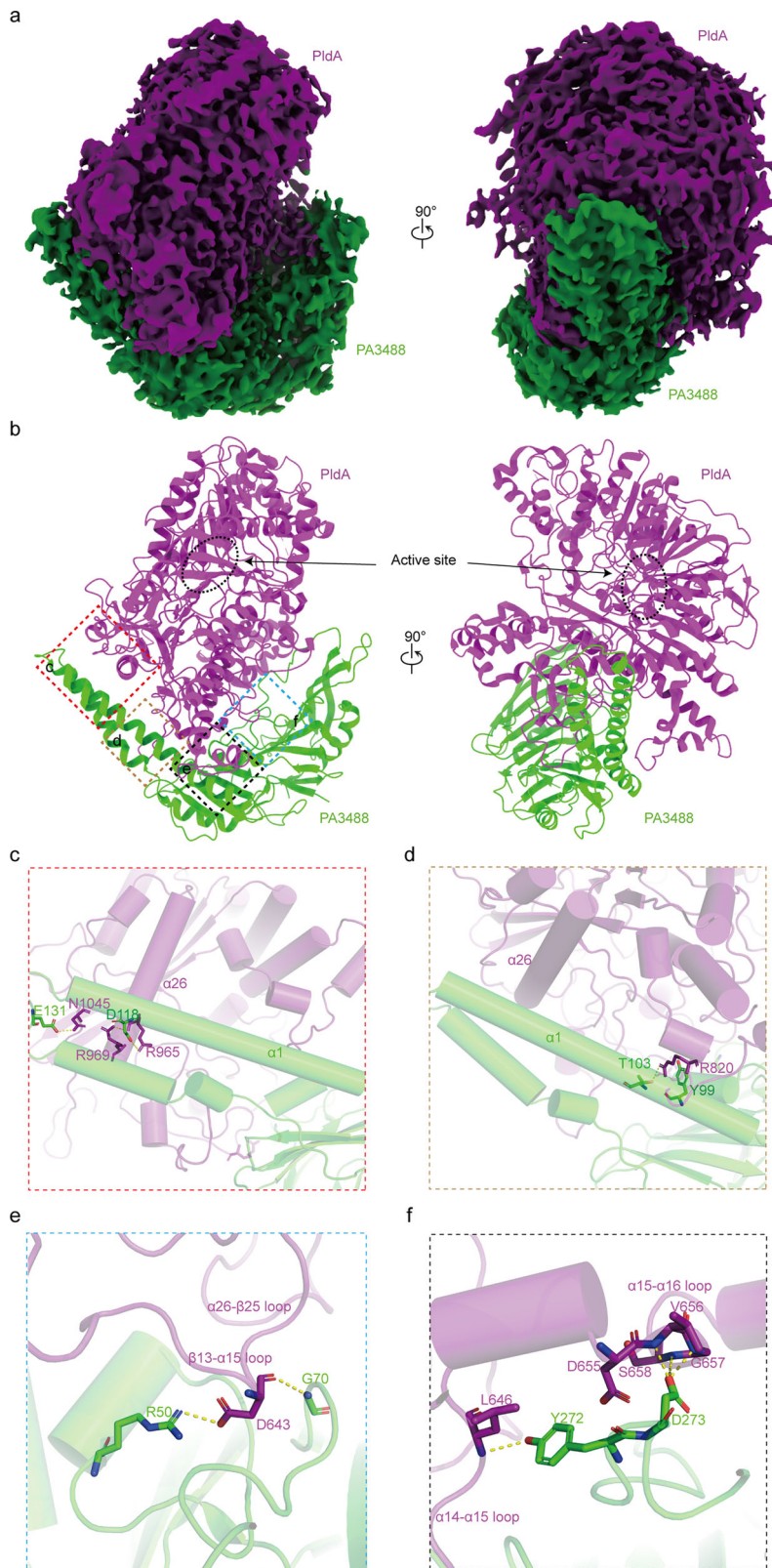

**Fig. 4 | Cryo-EM structure of PldA–PA3488 complex. a, b** Two orthogonal views of the cryo-EM density map (**a**) and model (**b**) of the PldA–PA3488 complex are shown: PldA and PA3488 are colored in purple and green, respectively. The dotted circle regions indicate the active site. **c**–**f** Details of the interaction between PldA and PA3488. The interaction residues are presented as sticks. Close-up view of interaction around PldA and C-terminal of α1 from PA3488 (**c**). Details of the interaction around PldA and N-terminal of α1 from PA3488 (**d**). Detailed interaction between PldA and central loops of PA3488 (**e**). Details of contacts around PldA and β13-β14 and β17-α4 loops of PA3488 (**f**).

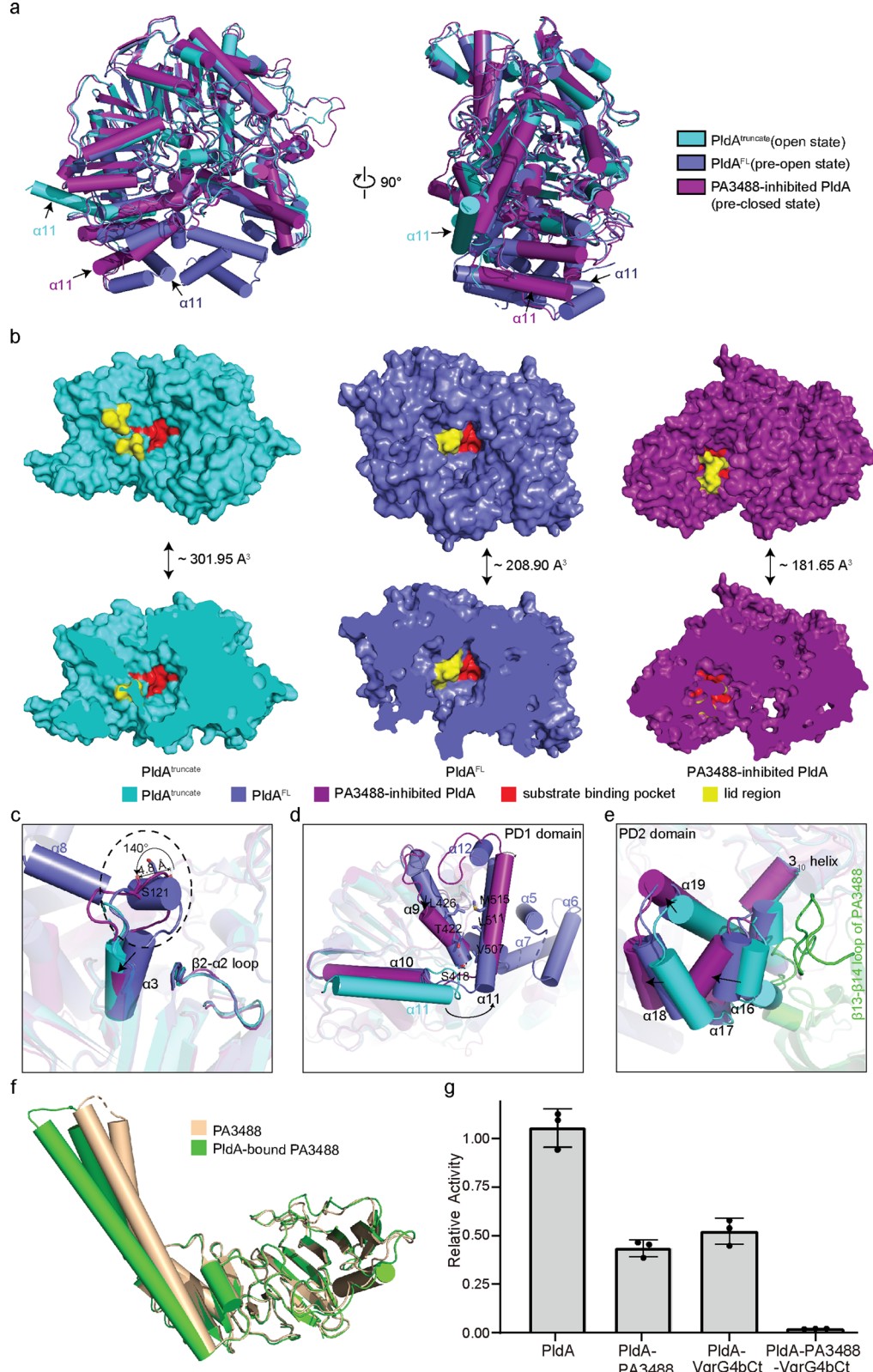

**Fig. 5 | PA3488 binding results in conformational changes in the structure of PldA and associated enzymatic assays. a** Structural comparison of PldA$^{truncate}$, PldA$^{FL}$ and PA3488-inhibited PldA. **b** Surface (up) and cross-sectional (down) representation of active pockets from PldA$^{truncate}$, PldA$^{FL}$ and PA3488-inhibited PldA. The lid region and the active site are highlighted in yellow and red respectively. The volume of each pocket in PLDs is calculated through a web server-CASTp 3.0 (http://sts.bioe.uic.edu/castp/). **c–e** Close-up views of the conformational changes in the lid region (**c**), PD1 (**d**), and PD2 (**e**). **f** Structural comparison of apo-PA3488 (PDB: 5XMG) and PldA-bound PA3488. **g** Inhibition of PldA activity by PA3488 and VgrG4bCt (aa 693–808). Results are means ± SD, $n = 3$ biologically independent samples. Dots indicate the corresponding data distribution. Source data are provided as a Source data file.

**Table 1 | Kinetics and affinity constants for wild-type and mutant PldA–PA3488 complex**

| Variants | Association rate Ka (1/Ms) | Dissociation rate Kd (1/s) | Binding affinity KD (M) |
|---|---|---|---|
| PldA | 5.96E + 05 | 1.83E−05 | 3.07E−11 |
| PldA$^{D643A}$ | 2.17E + 04 | 1.83E−05 | 8.46E−10 |
| PldA$^{R820A}$ | 2.13E + 03 | 5.19E−08 | 2.44E−11 |
| PldA$^{R969A}$ | 1.18E + 04 | 7.57E−05 | 6.42E−09 |
| PldA$^{D643A-R820A-R969A}$ | 2.40E + 03 | 1.78E−06 | 7.41E−10 |
| PA3488$^{R50A}$ | 2.08E + 05 | 2.14E−04 | 1.03E−09 |
| PA3488$^{D118A}$ | 3.30E + 05 | 6.25E−06 | 1.90E−11 |
| PA3488$^{D273A}$ | 6.92E + 03 | 1.20E−04 | 1.73E−08 |
| PA3488$^{R50A-D118A-D273A}$ | 2.71E + 05 | 3.82E−03 | 1.41E−08 |

Source data are provided in Supplementary Fig. 10.

outside when compared with the previously published apo PA3488[37] (Fig. 5f).

The resulting density map allowed us to unambiguously trace the interaction between PldA and PA3488. Four interfaces participate in the formation of the PldA–PA3488 complex (Fig. 4). First, the C-terminal of the long α1 helix from PA3488 interacts perpendicularly with the C-terminal part of the α26 helix in PldA, mediated by a pair of hydrogen bonds formed by the Glu$^{131}$ of PA3488 and the Asn$^{1045}$ of PldA, and a pair of salt bridges between the Asp$^{118}$ of PA3488 and the Arg$^{969}$ of PldA (Fig. 4c). In addition, the N-terminal of α1 in PA3488 makes contact with the loop connecting the α23 and β15 helices of PldA. In detail, the aliphatic side chain of the Arg$^{820}$ from the PldA stacks with the Try$^{99}$ aromatic ring of PA3488, anchoring the α26 helix and α23-β15 loop of PldA to the long α1 helix of PA3488 (Fig. 4d). Moreover, the β13-α15 and α26-β25 loops of PldA protrude into the concave surface of the central loops from the β3-β4 and β13-β14 loops of PA3488. The specificity of binding is due primarily to a salt bridge between the Asp$^{643}$ of PldA and Arg$^{50}$ of PA3488, combined with complementary van der Waals interactions between the Asp$^{643}$ of PldA and the Gly$^{70}$ of PA3488 (Fig. 4e). Finally, the α15-α16 and α19-α20 loops of PldA pack against the other side of PA3488 formed by the β13-β14 and β17-α4 loops to further strengthen the interaction between PldA and PA3488. In this region, a cluster of residues in PldA (Asp$^{655}$, Val$^{656}$, Gly$^{657}$, and Ser$^{658}$) mediate a network of hydrogen-bonding interactions with the Asp$^{273}$ in PldA, helping secure the α14-α15 loop of PldA on the surface of PA3488. In addition to these hydrogen-bonding interactions, the Leu$^{646}$ in PldA is in close proximity to and therefore stacks with the Tyr$^{272}$ aromatic side chain of PA3488 (Fig. 4f).

To validate the residue interactions observed in the PldA–PA3488 structure, we generated several mutants and measured their binding affinity by surface plasmon resonance (SPR). The results suggested that PldA$^{D643A}$ and PA3488$^{R50A}$ dramatically decrease the PldA–PA3488 interaction, which is consistent with their functions in stabilizing the PldA–PA3488 complex through forming a tight salt bridge (Fig. 4e, Table 1 and Supplementary Fig. 10). In addition, the PldA$^{R969A}$ and PA3488$^{D273A}$ mutants severely disrupted the PldA–PA3488 interaction, indicating that these two residues also function as the major structural determinants in PldA–PA3488 recognition (Fig. 4c, f, Table 1 and Supplementary Fig. 10).

### Conformational change of PldA

A comparison of the apo-PldA and PA3488-inhibited structures of PldA enables us to investigate the conformational changes that are associated with complex formation. Close inspection of the structures obtained in this study indicated that the conformational changes mainly occurred in the lid and the PD region (Fig. 5a–e).

In PldA$^{FL}$, the lid region forms a short $3_{10}$ helix (residues 120–122), packing with the β2-α2 loop and the α8 helix. In such case, the lid

region is positioned distant from the entrance of the active-site pocket, which becomes accessible to the solvent so that the substrate can gain access (Fig. 5a–c). In contrast, in the inhibited PldA, on account of the binding of PA3488, the $3_{10}$ helix melts, allowing the region to instead form an extended loop, stretching out and following the α3 helix direction and resulting in the formation of a shorter α3 helix than that of apo-PldA (Fig. 5a–c). This is associated with a translation of the lid toward the active site and a rotation of roughly 140° (based on Cα of Ser$^{121}$), so that the backbone atoms in the lid region are as much as 4.8 Å away from their counterparts in PldA$^{FL}$ (Fig. 5a–c). Conclusively, the rotation transforms the relative "phase" of the lid region in the two structures. On binding PA3488, the whole lid region slightly changes its position and orientation with respect to the active site, sitting on the top of it and almost entirely occluding the entrance of the solvent and substrate (Fig. 5b). Coincidently, activity assays showed that the PldA almost but not completely lost its enzyme activity upon PA3488 binding (Fig. 5g); hereafter, we refer to PA3488-inhibited PldA as "pre-closed" PldA.

The altered position and orientation of the lid region is directly coupled to the PD region. Of note, the α5-α8 loop of the PD1 domain in the inhibited PldA is structurally invisible in the electron microscopy map, implying that these helices are highly dynamic after PA3488 binding, and this accordingly imparts flexibility to the lid region. In PldA$^{FL}$, the lid region stacks against the α8 helix, creating a stabilized $3_{10}$ helix. In PA3488-bound PldA, the flexibility of α8 substantially relieves the steric hinderance, making space for the lid region to move toward the active site pocket and block it. The flexibility of α8 also influences the conformation of the following α9 helix, where the Leu$^{426}$, Thr$^{422}$, and Ser$^{418}$ pack extensively with the Met$^{515}$, Leu$^{511}$, and Vla$^{507}$ in the α11 helix of PldA$^{FL}$. On binding PA3488, the C-terminal side (residues 429-432) of α9 unravels into a loop and moves relative to the HKD1 domain, with concomitant displacement of the associated helices α11 and α12 (Fig. 5d). Meanwhile, the PD2 domain also moves slightly inside (Fig. 5e). In PldA$^{FL}$, the α15–α19 helices in the PD2 domain point in the opposite direction relative to the HKD2 domain. In PA3488-bound PldA, however, a long β13-β14 loop in PA3488 is incorporated into α15-α19 through five hydrogen bonds (with residues 646 and 655–658) (Fig. 4f). To accommodate the loop of PA3488, a $3_{10}$ helix (residues 656–658) of PldA unwinds, and its melting on binding PA3488 enables the adjacent helix α16 to move toward α18, tilting away from their positions in PldA$^{FL}$. The interconnected α17 and following α19 are also coupled to the changes that α16 and α18 undergo (Fig. 5e). Consequently, the movements of the lid region and the PD1 and PD2 domains are correlated.

In PldA$^{truncate}$, the only partially visible loop of the lid represents the fully open state of the substrate-binding pocket (Fig. 5a, c), and the entrance of the substrate-binding pocket is shallower and more open than those of PldA$^{FL}$ and inhibited PldA (Fig. 5b). Furthermore, as the first door shield, the PD region also further unbolts, accompanied by the disassembly of the PD1 and PD2 domains. This is consistent with the movement of the indicator α11 shifting 120° toward the outside (Figs. 3a, c and 5a–c).

In summary, these structures provide insights into the mechanisms by which PA3488 inhibits the toxicity of PldA by sequential conformational changes.

## Discussion

*P. aeruginosa*, a causative agent of hospital-acquired infections, secretes a eukaryotic-like protein, phospholipase D (PLD), known as PldA, which is responsible for its virulence[2,31,38]. PldA has extensive homology with the PLDs of eukaryotes, but not with those of prokaryotes[12]. Given its comparatively larger molecular weight of 122 kDa and its inherently disordered region, it has been challenging to perform a structural characterization of PldA. Here, we have determined, for the first time, the crystal structure of the full-length

eukaryotic-like PldA (PldA^FL) in the apo state. The high-resolution structure that we present here allows illustration of the similarities and differences in structure and function between PldA^FL and PLDs from different species (bacterial PLD, human PLD, and plant PLDα). Indeed, conserved structural features include the canonical "horse saddle" topology, the two HKD domains, and the active site in which residues involved in catalysis are located. Mutagenesis studies confirmed again the importance of these critical catalytic residues (Fig. 1e). These similarities support the conservation of the mechanisms of substrate binding and catalysis in both bacterial and eukaryotic PLDs.

PldA also exhibits some structural features that are distinct from other PLDs. First, a PD region, which does not exist in other PLDs is situated adjacent to the catalytic domain, possibly imparting unique characteristics to the active site in PldA. This arrangement of helices, combined with the lining of the hydrophobic residues inside the helices, makes it easy to hypothesize that these helices may play an important role in regulating enzymatic activity. Human PLDs possess tandem phox homology (PX) and pleckstrin homology (PH) domains, which are known to mediate interactions with lipid membranes and are believed to regulate PLD localization within the cell. In view of the significant homology between PldA and hPLDs, it is tempting to propose that the PD region may have functional links with the PX–PH domain of hPLDs. Defining the nature of this region involved in binding of the substrate and lipid membrane, as well as characterizing the physiological relevance with the PX–PH domain of hPLDs, may be critical for understanding the mechanism of infection of PldA.

In addition, the core catalytic domain of PldA^FL has much more in common with hPLDs than with plant PLDα (Supplementary Fig. 2b). One main feature makes PldA and hPLDs different from their plant cognate: the "open" or "closed" state generated by the position and orientation of the lid region. Our structures of PldA^FL and the existing hPLD structures show that they adopt an "open" conformation, with the potential lid region draping away from the active site and allowing the accessibility of the active site to the solvent and the substrate. In plant PLDα, the reverse applies: in the native PLDα structure, the lid hinders the access of water and the substrate to the active site, while the binding of reaction product PA8 triggers the conformational change of the lid region to an open state, making the active site accessible to both solvent and substrate[24]. One pioneering crystallographic study on hPLD1 described the possibility of occluding the active site by auto-inhibition[23]. This study assumed that the two amino acids (Trp^381 and Arg^917) may inhibit the entrance of the substrate due to their position relative to the active site. By analogy with the either "open" or "closed" structure[24] of plant PLDα and our structure of PldA, we suggest that these residues rarely contribute to the state of the catalytic pocket, as they adopt the same conformation in these structures. It is unknown whether PldA and hPLDs readily interconvert between "open" and "closed" conformations, as observed for plant PLDα. However, in this study, we have also determined the crystal structure of PldA using in situ proteolysis (named PldA^truncate) and the cryo-EM structure of PldA in complex with PA3488–an immunity protein that can suppress the virulence of PldA. Compared to the open conformation exhibited by PldA^FL (pre-open state), PldA^truncate presents a highly open conformation (open state) and the PA3488-inhibited PldA reveals an almost closed conformation (pre-closed state). The transition between these conformations involves the realignment of the PD domain and lid regions. The concerted movements of these two regions eventually affect their positions and orientations relative to the active site and hence free or obstruct the entrance of the solvent and substrate. These observations indicate that PldA may undergo large conformational changes in response to substrate binding.

Based on the structures obtained in this study, we propose the following mechanism for the invasion of PldA into a target cell. Inside *P. aeruginosa* cells, the PD2 of PldA interacts with PA3488, and an unknown chaperone protein interacts with PD1 of PldA to stabilize and tightly close the PD region and active pocket together (closed state). Once receiving a physiological signal to secrete PldA to adjacent cells, PA3488 and an unknown protein will be released from PldA to open the PD door shield, and the active pocket gradually opens (pre-closed and pre-open states). Upon anchoring the cell membrane of the target cell, the PD domain and active pocket are fully open to degrade phospholipids efficiently (open state) (Fig. 6). It has been reported that the VgrG proteins are encoded adjacent to PldA and can directly interact with it[31]. Recently, effectors linked by VgrG have been reported to require effector-specific chaperones for stability and/or to favor their interaction with VgrG[39]. Therefore, it is possible that there are unknown chaperones that can work with PA3488 to "lock" PldA in the closed state. Indeed, we found the coexistence of VgrG4b (PA3486) and PA3488 can completely inactivate enzymatic activity of PldA (Fig. 5g), as leakage of this enzyme activity would cause cell death by self-toxicity.

There is increasing evidence that PldA is associated with promoting chronic infection and targeting eukaryotic cells to produce cell invasion. Given its sequence and particularly its structural similarities, one can easily speculate that PldA is capable of mimicking certain functions of the host PLD; this may partly account for its mechanism of activity as a virulence factor. The crystal structure of PldA^FL, PldA^truncate, and the cryo-EM structure of the PldA–PA3488 complex we present here thus provide a structural basis that sheds light on the activation and inhibition mechanisms of PldA. This should aid the future design of PLD-targeted inhibitors and drugs.

## Methods

### Protein expression and purification

*P. aeruginosa* PldA^FL and PldA^truncate were expressed in *E. coli* Rosetta cells using a modified pGEX-6t vector with an N-terminal 6×His tag prior to the GST fusion tag that is removable by cleavage with tobacco etch virus (TEV) protease. The cells containing expression plasmid were induced at $OD_{600} = 0.6$ for 16 h with 0.1 mM isopropyl-β-D-1-thiogalactopyranoside at 16 °C, and the cells were harvested by centrifugation. The cells were then resuspended in lysis buffer (20 mM Tris pH 7.5, 500 mM NaCl, 5% Glycerol, and 1 mM phenylmethylsulfonyl fluoride) and lysed by sonication. Cell lysate was cleared by ultracentrifugation at $15,000 \times g$ for 40 min, and the supernatant was incubated with nickel-nitrilotriacetic acid resin (Bio-Rad) at 4 °C for 30 min. After washing, the bound proteins were eluted by 300 mM imidazole, followed by hydrolysis with TEV protease to remove the 6× His-GST tag and reloaded with 20 mM imidazole. The protein sample was further purified by gel-filtration chromatography (Superdex 200, GE Healthcare) equilibrated with a buffer (20 mM Tris pH 7.5, 150 mM NaCl). The purified proteins were concentrated to 10 mg/mL and stored at −80 °C. *P. aeruginosa* PA3488 (20–376, without the N-terminal 19 signal peptide) was expressed as stated above, PA3488 and PldA^FL were co-purified using the same protocols for the PldA. The purified PA3488-PldA complex were concentrated to 10 mg/mL and stored at −80 °C. Mutations were produced by PCR-based site-directed mutagenesis, and the designed mutated proteins were purified using the same strategy as described above.

### Crystallization and structure determination

Crystallization screening of PldA^FL and PldA^truncate was carried out at 20 °C using the sitting-drop vapor-diffusion technique. The best crystals of the native PldA^FL were grown under the conditions of 0.2 M potassium chloride, 0.05 M magnesium chloride hexahydrate, 0.05 M Tris hydrochloride pH 7.5, and 10% polyethylene glycol 4000. The crystals of PldA^truncate were obtained by adding chymotrypsin to the crystallization mixture in a ratio of 1:100; in this way the enzyme acts on the PldA^FL under crystallization conditions and allows the proteolytic fragment to form crystals in the same drop. The best crystals of the PldA^truncate were grown under the conditions of 0.2 M potassium

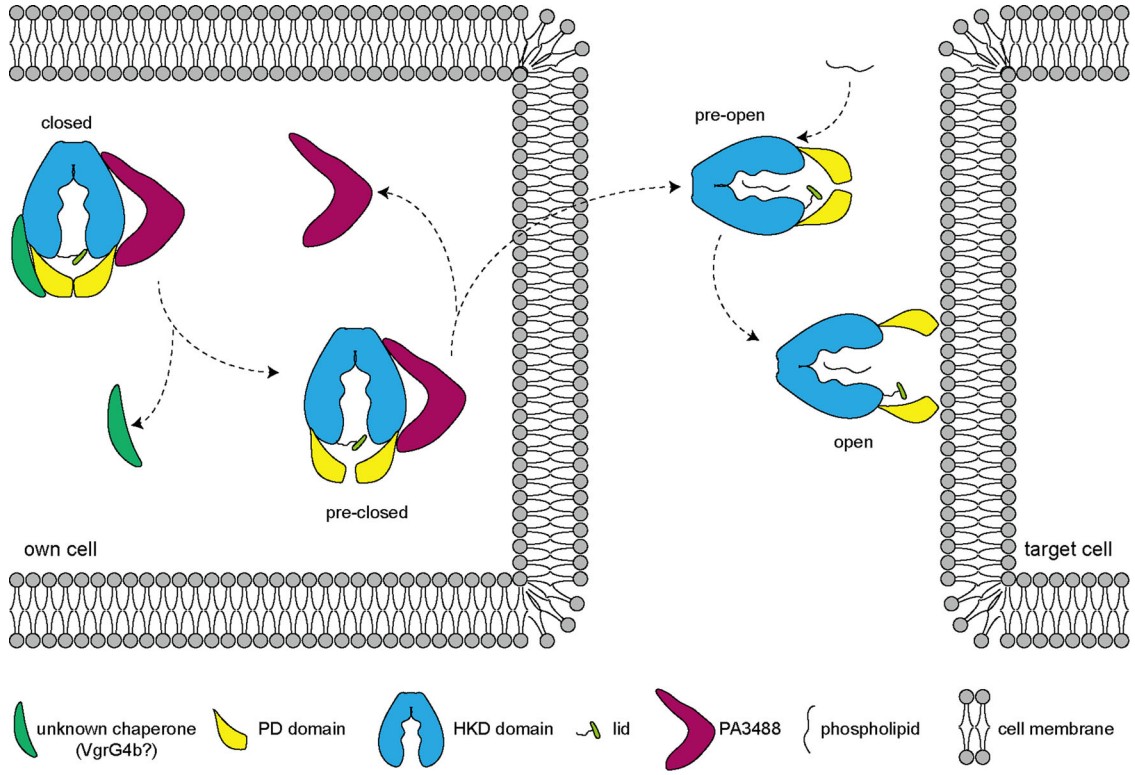

**Fig. 6 | Mechanism of PldA-mediated invasion.** Proposed model for the invasion of PldA into target cell. PA3488 and an unknown chaperone (probably VgrG4b) interact with PldA to tightly close the active site (closed state) inside their own cells; once receiving a physiological signal to secrete PldA to adjacent cells, PA3488 and an unknown chaperone is released from PldA to open the PD door shield and the active pocket gradually opens (pre-closed and pre-open states); upon anchoring the cell membrane of the target cell, the PD region and active pocket are fully open to degrade phospholipids efficiently.

phosphate dibasic pH 9 and 20% polyethylene glycol 3350. Datasets of PldA^FL were collected at 100 K on the BL19U1 beamline of the Shanghai Synchrotron Radiation Facility (SSRF), and datasets of PldA^truncate were collected at 100 K on the BL17U beamline of the SSRF. A 2.0 Å native dataset of the PldA^FL and a 3.0 Å native dataset of the PldA^truncate were processed with the XDS software package[40].

Attempts have been made to solve the phase problem via soaking the crystals in various buffers containing heavy metals, such as Pt, Hg, and Au, and incorporating selenium in the form of selenomethionine, but these were unsuccessful. Ultimately, despite the low sequence homology of PldA with PLDα (23% identity), the PldA^FL structure was determined through molecular replacement using PHASER from the PHENIX software package[41]. The structure of the *A. thaliana* PLDα (PDB code: 6KZ9) was used as the search model. The model was further refined using PHENIX and manual rebuilding in COOT[42]. The PldA^truncate structure was solved by molecular replacement using PHASER with PldA^FL as the search model. PHENIX and COOT were used repeatedly for refinement and manual building. All structural figures were prepared using UCSF Chimera[43], Chimera X[44], GraphPad Prism 5 (https://www.graphpad.com/), and PyMOL (https://www.pymol.org). All the data collection and structure refinement statistics of PldA^FL and PldA-^truncate are summarized in Supplementary Table 1.

## PldA activity assay
The assay was performed using the commercial Amplex Red Phospholipase D Assay Kit (Invitrogen, A12219) to analyze the enzyme activity of PldA in vitro. The assay was carried out at 37 °C for 10 min containing 0.2 µg of PldA, 125 µM di8:0 PC (dissolved in 20 mM Tris-HCl, pH 8.0, 150 mM NaCl) and the supplied regents of the kit. The fluorescence was excited at 530 nm and detected at 590 nm. For the assay of PldA inhibition by PA3488 and VgrG4bCt (aa 693–808), The molar ratio of the samples are: PldA: PA3488 = 1:1, PldA: VgrG4bCt = 1:1,

and PldA: PA3488: VgrG4bCt = 1:1:1, repectively. Kinetic study of PldA^FL and PldA^truncate activity were performed under standard assay conditions with various substrate concentrations, ranging from 1 to 200 µM di8:0 PC.

## Molecular docking
Molecular docking was performed using the AutoDock 4 software package[45]. AutoDockTools was used to generate the PA8 molecule and produce the area for docking. Ten states for the PA8 docking were generated, and the model with the lowest binding energy was selected for analysis. The images of these structures were generated by PyMOL.

## Molecular dynamics simulations
Molecular dynamics simulations of PldA^FL were performed using the GROMACS package[46] with the CHARMM36 force field. The protein was centered in a cube and dissolved with TIP3P waters. Sodium and chloride ions were added to neutralize the electric charge. The particle mesh Ewald method was used to compute the electrostatic interactions with a real-space cut-off distance of 1 nm, and the van der Waals interactions were set at the same cutoff value. The system was energy minimized and heated to 300 K before the production process. The trajectory production processes proceeded for 70 ns, and the root-mean-square deviations and fluctuations were calculated on the Cα atoms of each residue.

## Cryo-EM sample preparation and single-particle dataset acquisition
4 µL of PldA-3488 complex was applied onto a plasma-cleaned R1.2/1.3 300-mesh Au holey-carbon grid (Quantifoil), blotted using a Vitrobot Mark IV (Thermo Fisher Scientific), and plunge frozen in liquid ethane (15 s wait time, 8 °C, 100% humidity, 3–6 s blot time, −1 blot force). The sample quality was screened using a Titan Krios electron microscope

equipped with a Falcon III direct electron detector (Thermo Fisher Scientific). Samples with good quality were imaged on the same electron microscope at 300 kV, equipped with a GIF Quantum energy filter (the slit width was set to 15 eV) and a K3 Summit direct electron detector (Gatan) in the super-resolution counting mode. A total of 3732 movie stacks were collected using SerialEM[47] at a magnification of 130,000. During data collection, the defocus value was set in the range from 1.5 to 2.0 μm. Each movie stack was dose-fractioned into 32 frames with a total dose of about 50 e⁻ Å⁻². More data collection details are given in Supplementary Table 2.

## Image processing
All movie stacks were motion-corrected, binned two-fold (0.668-Å pixel size after binning) and dose-weighted using the MotionCor2 program[48]. Contrast transfer function (CTF) estimation was carried out using the CTFFIND4 program[49]. The following image processing steps were carried out in RELION 3.1[50], as shown in fig. S4. A total of 2,099,492 particles with diameters ranging from 10 to 14 nm were auto-picked by a blob picker, and these were randomized split into four subsets. Each subset was subjected to several rounds of 2D classification, and particles with ice contamination or low resolution were removed. A total of 12 good classes with 1000 particles per class were selected to generate an initial model without reference. All selected particles (1,158,720) were combined and subjected to a 100 iterations global angular searching 3D classification with only one class, which to make the initial angle information of each particle more accurate and more reliable. For each of the last three iterations of the 3D classification above, a 25 iterations local angular searching 3D classification was performed with 10 classes. Good classes from each local angular searching 3D classification were selected and merged. The duplicated particles were removed, resulting in 577,905 particles in total. After 3D auto-refinement without any symmetry imposed, these particles yielded a reconstruction at an overall resolution of 3.32 Å. After CTF refinement and post-processing, the map resolution was improved to 3.05 Å. All resolutions were estimated using two independently refined half maps[50] with the gold-standard Fourier shell correlation (FSC) = 0.143 criteria[51]. Directional FSCs were calculated using the 3DFSC server[52], and local resolutions were determined with RELION's own implementation.

## Model building, refinement, and validation
The 3.05 Å cryo-EM map was used for model building. The crystal structures of PldA and PA3488 (PDB:5XMG) were fitted into the cryo-EM map using UCSF Chimera[43], and these were used as the guide to build the model. The model was checked and corrected manually using COOT[42], and it was refined against the map using PHENIX[41] with secondary structure restraints applied. For model cross-validation, the coordinates of the final model were randomly shifted by up to 0.5 Å and then refined in PHENIX against one of the unfiltered half-maps. The refined model was then tested against the other unfiltered half-map. The model statistics are shown in Supplementary Table 2.

## SPR experiments
The interactions between PldA and PA3488 were explored using a Biacore 8 K (GE Healthcare) instrument against immobilized PldA on the sensors at 298 K. The PldA samples were diluted to 80 nM in 25 mM HEPES, 250 mM NaCl, 0.05% (v/v) Tween 20, pH 7.2, and were immobilized on the flow cells of a Biacore CM chip (GE Healthcare) to 12295, 12500, and 11925 resonance units. Gradient concentrations of PA3488, mutant protein PA3488[R50A], PA3488[D118A], PA3488[D273A], and PA3488[R50A/D118A/D273A] (from 100 nM to 6.25 nM with two-fold dilution) were then flowed over the chip's surface. After each cycle, the sensor surface was regenerated with Gly-HCl pH 1.7. The binding kinetics were analyzed using a 1:1 binding model with the BIAevaluation software package (GE

Healthcare). In turn, the binding of PldA and its mutants—PldA[D643A], PldA[R820A], PldA[R969A], and PldA[D643A+R820A+R969A]—with PA3488 were also measured when immobilized by PA3488 on the chip following the same method as above.

## μ-XRF data acquisition
For synchrotron micro X-ray fluorescence (μ-XRF) analysis, 100 μL purified PldA protein (15 mg/mL, in buffer containing 20 mM Tris-HCl, pH 8.0 and 150 mM NaCl) was mounted onto Kapton tape. 100 μL 0.1 mM CaCl₂ was also analyzed as a reference. The distribution of Ca was measured at the 4W1B beamline in Beijing Synchrotron Radiation Facility (BSRF, China). The electron energy in the storage ring was 2.5 GeV with a current ranging from 200 to 300 mA. The incident beam was focused with a size of 50 μm × 50 μm. A monochromatic X-ray with a photon energy of 15 keV was used to excite the samples and the count time was 10 s per pixel. PyMca software and origin 8.0 were used to process the data and plot the elemental distribution of PldA protein, respectively.

## MALDI-TOF MS sample preparation and data processing
The crystals of PldAtruncate were collected, washed by crystallization buffer and dissolved in 6 μl H₂O. The sample was then visualized by 12% SDS-PAGE gel. The protein bands were excised from the SDS-PAGE gel. After dehydration, the gel plugs were incubated in 25 mM NH4HCO₃ with 5 mM DTT for 45 min. Then, the samples were alkylated with 40 mM iodoacetamide in 25 mM NH₄HCO₃ for 45 min at room temperature in the dark and digested overnight with trypsin (40 ng for each band) at 37 °C. The reactions were terminated by adding trifluoroacetic acid to a final concentration of 1%, and desalted using C18 Zip-Tip microcolumns (Millipore, Germany). The samples were then loaded onto the instrument in a crystalline matrix of α-cyano-4-hydroxycinnamic acid (CHCA, 5 mg mL⁻¹).

MALDI-TOF/TOF-MS analysis was performed on an UltrafleXtreme mass spectrometer controlled by FlexControl 3.4 software package (Bruker Daltonics, Bremen, Germany). The instrument was externally calibrated using the Bruker peptides calibration kit. The spectra were acquired in the positive ion reflectron mode over the $m/z$ range from 700 to 3500. For the MALDI-TOF/TOF-MS analysis, precursors were accelerated and selected in a time ion gate, after which fragments arising from metastable decay were further accelerated in the LIFT cell and detected after passing the ion reflector. The MALDI-TOF/TOF MS spectrum was subjected to a database search using the Mascot v2.5 (Matrix Science, London, UK) database search engine (www.matrixscience.com). The search parameters were as follows: enzyme: trypsin, allow up to 2 missed cleavages, fixed modifications: Carbamidomethyl (C) and variable modifications: Oxidation (M). The sequence of PldA was used for search database.

## Reporting summary
Further information on research design is available in the Nature Research Reporting Summary linked to this article.

# Data availability
The coordinates and structure factors for PldA[FL] and PldA[truncate] have been deposited in the Protein Data Bank with accession codes 7V53 and 7V55, respectively. The cryo-EM 3D maps of the PldA–PA3488 complex have been deposited in the Electron Microscopy Data Bank (EMDB) with accession code EMD-32438. The original data for single particle analysis of PldA–PA3488 complex can be downloaded from EMPAIR. Atomic coordinates of PldA–PA3488 complex PDB: 7WDK have been deposited in the Protein Data Bank. All data needed to evaluate the conclusions in the paper are shown in the article and/or the supplementary materials. Source data are provided with this paper.

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

## Acknowledgements

We are grateful to the staff members of SSRF for sample test and data collection. We sincerely thank the staff at both the cryo-EM center of South University of Science and Technology and the Tsinghua University Branch of the National Protein Science Facility (Beijing) for their technical support on the Cryo-EM and High-Performance Computation platforms. This study was financially supported by the grants from the National Basic Research Program of China (2017YFA0504900), National Natural Science Foundation of China (Grant number 31700651), Beijing Municipal Science & Technology Commission (No. Z191100007219007), and the Strategic Priority Research Program of CAS (XDB37040302).

## Author contributions

Y.L., Y.D., and S.-F.S. supervised the project; X.Y. and Z.L. carried out protein purification, crystallization experiments, and performed the biochemical analysis; L.Z. and Z.L. collected the EM data and performed the EM analysis; Z.G. collected and analyzed X-ray data; X.Y., Z.L., Z.S., and Y.L. prepared the manuscript; Y.D. and S.-F.S. edited the manuscript.

## Competing interests

The authors declare no competing interests.
