## [Peer Review File · Nature Communications]

Structural insights into PA3488-mediated inactivation of *Pseudomonas aeruginosa* PldAReviewer #1 (Remarks to the Author):

Pseudomonas aeruginosa is an opportunistic bacterial pathogen that contains many virulence-associated genes. One of the genes is a phospholipase D enzyme called PldA that is more similar to eukaryotic PLDs than bacterial PLDs. Yang et al report the first high resolution crystal structure of Pseudomonas aeruginosa PldA that reveals a common PLD fold with a eukaryotic-like C-terminal domain, and a unique peripheral helical domain (PD domain) that appears to sit above the active site. A crystal structure of a truncated PldA is also determined. In this truncated structure, the catalytic domain is open to solvent with most of the flexible PD domain seemingly missing electron density, as well as a lid region in PldA. The authors next determine the structure of PldA bound to PA3488 using single particle cryoelectron microscopy. PA3488 binds adjacent to the catalytic domain and causes conformational changes in the lid region. PA3488 binds extremely tightly to PldA and mutations at the interface reduce the affinity of this interaction. Activity assays using a short chain lipid substrate show that in the presence of PA3488A the activity of PldA is reduced but not eliminated. Based on this structural work, a final model is proposed where PldA is initially bound to PA3488 and an unknown chaperone to inhibit PldA activity in a closed conformation, then secreted into a pre-open conformation, and finally adopting an open conformation that is a fully active PLD.

Overall the structure of this eukaryotic-like bacterial PLD is very interesting and there is a nice comparison with the eukaryotic PLDs in the discussion. However, some aspects are not clear and could be improved with modifications to the figures/text and additional experiments. For example, clarification of the biochemical assays and additional experiments would help support the proposed model, especially with regard to inhibition by PA3488 and the proposed increased activity in the open structure of PldA-truncated. Also, for conformational changes it is unclear if the conformational differences in the PldA-truncated structure are simply due to the lower resolution and lack of density for the PD region. Lastly, the exact roles of PldA and PA3488 in P. aeruginosa infection and virulence are unclear in some sentences.

Points to address

- 1. It is stated that PA3488 is an immunity protein of PldA that can neutralize the virulence of PldA. However there is no reference for this statement, so it is unclear what evidence there is and how PA3488 impacts virulence.**
- 2. In the discussion, lines 289-290. It is stated that PldA is essential for virulence, but no reference is given. Previously (lines 88-92) it was stated that it has roles in inter-bacterial competition, eukaryotic host infection, and mammalian cell invasion. Does genetic deletion of PldA cause of loss of virulence? If not, the statement above should be toned back.**
- 3. The authors describe the C-terminal domain as very similar to the eukaryotic human and plant PLD C-termini. However, this is not easily visualized in figure in 1c. Can this be more explicitly shown in a supplemental figure or new panel?**
- 4. What regions of truncated PldA have been removed? Is loss of these regions the reason for the observed structural differences? Or are the structural differences due to the lower resolution and lack of electron density for the PD region?**
- 5. The structure of truncated PldA has a calcium ion bound (table 1), and looking at the literature there is evidence that calcium increases the activity of PldA (see <https://doi.org/10.1046/j.1365-2958.2001.02282.x>). Does calcium increase the activity of PldA in the authors hands too? Was calcium included in the crystallization conditions? Where does this calcium bind and is the coordination consistent with calcium? How was this metal identified as calcium? Lastly, could the presence of calcium affect the conformation of PldA and its activity?**

6. The proposed model uses the open conformation of truncated PldA as a rationale for the proposed conformational changes. How does the activity of truncated PldA compare to the full-length protein? And is this consistent with the model?

6. It is stated that PA3488 is an inhibitor/chaperone of PldA, however PA3488 does not fully inhibit PldA (extended data Fig. 3C) with ~40% activity estimated to still remain. How do the authors rationalize this given the structural observations of the closed state.

7. Looking through the literature there was a previous, quite thorough, biochemical characterization of PldA activity by Spencer and Brown (DOI: 10.1021/bi501291t) using a liposome system, rather than short chain lipids. However, I do not see a reference to this work. From reading that paper, it seems PldA has a preference for phosphatidylethanolamine (PE) >>> phosphatidylcholine (PC), but can also hydrolyze all phospholipids except PI.

In this study, it is unclear how the activity assay was performed. The methods states that a short chain phosphatidic acid was used as a substrate (di8:0 PA). However, PA is the product, so I assume this must have been a typo. Was short chain PC used? Or another phospholipid? I assume PC since the amplex red assay was used to detect production of choline? How was "PC" solubilized? E.g. by BSA, liposomes, micelles, or just aqueous solvent since the acyl chains are short. More details should be included in the methods section and figure legends.

Given the prior literature, it seems most relevant to use PE as a substrate than PC. Another major concern is the use of a short chain lipid, especially given the lack of complete inhibition by PA3488. It seems more relevant to use a liposome system for activity assays to better reflect a true membrane environment and the effect of PA3488 on PldA activity and potentially membrane binding.

8. In the cryoEM structure figure 4, it looks like there is a region modeled in the upper right panel (Fig. 4a) that does not have any cryo-EM density (right side, Fig. 4b). Is that correct? Or are the orientations different? It looks like this region corresponds to the PD region.

9. It would be useful to highlight where the active site of PldA is, in relation to bound PA3488.

10. It is stated that the full length PLD crystals had poor diffraction, so the truncated structure of PldA was determined. But the truncated structure is at a lower resolution. This seems opposite.

Reviewer #2 (Remarks to the Author):

Dear authors,

with great interest, I read your manuscript entitled "Structural insights into PA3488-mediated inactivation of *Pseudomonas aeruginosa* PldA". The manuscript presents two crystal structures of phospholipase D of *P. aeruginosa*, a full-length one and its truncated version, together with its structure bound to the PA3488 immunity protein, which was determined by cryo-electron microscopy. These three structures differ, highlighting the effects of the missing amino-acid chain and the PA3488 binding. Together with the knowledge of two phospholipid acid structures of *A. thaliana*, you proposed a model of how phospholipase D facilitates entry of *P. aeruginosa* pathogen into target cells.

In general, I believe that the presented research is solid and used appropriate materials and methods to draw correct conclusions, which will be valuable for a broad range of readers and important in the fight with *Pseudomonas aeruginosa*. The manuscript is also

written with good stylistics and has a very nice introduction section. However, in the current form it suffers from a large amount of ambiguities and requires thus a major revision.

Major points:

1. Starting model for model building of PldA: thank you for mentioning the difficulties with x-ray structure determination and molecular replacement. Did you try to confirm the resolved PldA structures with Alphafold PLdA structure of *P. aeruginosa* as a starting model?

2. CryoEM structure determination: What was your motivation to use data from all the last three iterations of local-search 3D classification and combine them later, discarding duplicates? Was it different from the results of the last iteration only? How many iterations were used in global and local runs? Please update the methods section accordingly. Which structure was used as initial 3D model? Could Bayesian polishing followed by another round of CTF refinement, or further 3D classification improve the map quality?

3. It is totally unclear which amino-acids were removed from PldA to form the truncated version and with which motivation. In the current state, it is impossible to interpret the truncated structure and its differences to the full-length counterpart. Please issue this information upon the first mention of the truncated version.

4. Inspecting the submitted pdb files, I noted a couple of unmodeled regions in all structures. Please, disclose this essential information. In particular: why are these residues missing in the cryoEM model and the full-length model? In particular, could AA 239-416 in cryoem structure, AA 396-416 in full-length structure could play a role in the structure? Please comment.

5. I am missing a figure showing the model fitted to the cryoEM density in the region of the active residues of the catalytic site and the PldA-PA3488 interface. Please add such figure.

6. In general, the manuscript suffers from the lack of correct referencing and unique and consequent shorname nomenclature. For a non-expert reader, it is difficult to discriminate among PldA, PID alpha, hPLD, etc. Please provide standard species naming upon first mention in the manuscript.

7. Together with point 6, it is essential to provide PDB codes or references for all structures mentioned. Which structure was used to model the PA3488 protein?

8. "PA8" misses a reference. I assume it is phosphatidic acid (PA stands already in the introduction, indeed). What is the "8"? Could you please compare the position of the docked PA in *A. thaliana* structure with your full-length and cryoem structures?

9. I find the structural differences between *A. thaliana* and *P. aeruginosa* "lid" regions rather large, and would hesitate to join them into one common mechanism. I suggest to refer to each "closed" or "open" structure separately rather than to "pre-open" and "pre-closed", and compare the two species PldAs afterwards. Could you please also comment on sequence similarity in the lid region?

Minor points:

1. Figure legends are in general very brief, please update them with PDB codes to the structures shown.

2. Legend text to Ext. Figure 3C is missing.

3. Figures also typically lack descriptors and amino acid coding, which makes it impossible to understand them. On top of it, I would suggest to switch to ribbon representation if structural differences are depicted, or compare the structures side by side (e.g. Fig. 1C, 2B, 5A). Please, consider also to enlarge the field of view a bit so that the reader can get more context information.

4. Introduction, p. 2, l. 58: please name some of the diseases

5. Results, p. 4., l. 107-110: please include the AA ranges for each domain, and depict the core catalytic domain in Fig. 1A.

6. Results, p. 4., l. 120: Which B-factor is meant?

7. Results, Eukaryotic-like catalytic core of PldA: It would be nice to state that both HKDs co-form to it right at the beginning.

8. Results, p. 5, l. 156: add reference to the corresponding figure.

9. Results, Substrate binding pocket: Please indicate AA involved.

10. Results, p. 5, l. 160: bPLD - please indicate species.

11. Results, p. 7, l. 210-211: mention AA involved in the lid. Fig. 4a doesn't show it.

12. Discussion, p. 10, l. 319-321: please include reference articles.

13. Fig. 2: describe water molecules and bond types in the legend.

14. Ex. Fig.1 b: color code missing. d: indicate position of PD helices in the figure.

With best regards!

Reviewer #3 (Remarks to the Author):

The authors describe an interesting study regarding the *P. aeruginosa* phospholipase D, PldA. The combined crystallographic/cryo-EM analysis of PldA with and without its immunity protein PA3488 yielded insights into the activation and inhibition of this protein. Although the study is quite interesting and the structural biology aspects of this study are adequately performed, there are many other serious issues that need to be addressed and a major revision is therefore requested.

Major shortcomings:

-Calcium is known to activate some bacterial phospholipase Ds, briefly mentioned on line 83. (but also PMC3233269); the authors did not probe this in their assays and should have performed such a study. Also, there is one Ca²⁺ in the 2nd crystal structure (truncated) yet the position of this calcium is not described. What is the role of Ca²⁺ in PLD activity? How does the activity of PLD depend on the concentration of Ca²⁺, where none of these catalytic residues (HKD) show binding of Ca²⁺? Does Ca²⁺ support structural stabilization of PLD?

-the catalytic activity of the truncated protein (obtained by limited proteolysis) was never measured and is needed for this structure-function study.

-the authors bring up that there is an unknown chaperone but do not provide any evidence (from the literature and/or is it in their cryo-EM structure?).

-the raw SPR data for the curves is not presented (only final K_a and K_d values are listed).

-there need to be replicates in both the SPR measurements and activity measurements to improve scientific rigor (p-values, standard deviations).

-In the discussion section, the authors compare the different PLDs with some domains being present and others absent. Although surface representations of these proteins are included in Extended data Fig. 2, it would be helpful if also cartoon representations of these proteins are shown with all the different domains colored consistently and the orientations are the same so the reader can follow the discussion.

-it would be informative if a Mass Spectrometry analysis were done on the crystals of the truncated protein to figure out the precise starting and ending amino acid.

-Did the authors check the crystal contacts, while comparing the movement of the α 11-helix as well other conformational changes to rule out the possibility of crystallization artifact of PldA(FL) and PldA(truncated) proteins in two different space groups?

-How the molecular surfaces and substrate-binding pockets of PldA, hPLD2, PLD α , and bPLD were compared? No surface cavity calculations were performed.

-The authors mention, the binding of PA3488 inactivates PldA, however in "Extended Data Fig. 3", panel C, the PldA+PA3488 shows partial activity. The authors should provide an explanation as the affinity between the proteins is less than 0.1 nM.

Minor comments:

-line 46: replace "helices" with "helical"

-line 46: replace "mutation assay" with "mutational analysis"

-line 46: replace "Otherwise" with "In contrast"

-line 77: "two HKD motifs located in the C-terminal catalytic domain". HKD motifs located well in the middle of the sequence.

-line 85: it is not clear what the authors mean. Perhaps it is better to replace "low" with "some detectable" if that is what the authors mean to convey (i.e., PldA does not have any detectable sequence identity? Or do the authors mean that PldA has high sequence identity).

-line 114: replace "almost same as" with "very similar to"

-line 119: replace "periphery" with "peripheral"

-line 122: replace "these" with "the positions of these". Also, delete the word "integrally"

-line 135: add "of wt activity" after "30%"

-line 137: add the word "region" after "C-terminal"

-line 143: hydroxy should be hydroxyl

-line 145: "binding" should be "bonding"

-line 150: replace "well" with "site"

-line 152: add "site" after "active"

-line 153: "configuration" should be "conformation"

-line 170: replace "against" with "at"

-line 192: PA8 is never defined, add a definition as people outside this field do not know what that is.

-line 198: replace "near" with "nearby"

-line 222: replace "stretches" with a different word

-line 338: replace "dominant" with "large"

-line 661: PldAD643+AR820+AR969A. Please correct the mutants R820A and R969A

-Extended Data Fig. 3- Description of 'c' panel missing.

Thank you for your letter and for the reviews' valuable comments concerning our manuscript (NCOMMS-22-05619-T). These comments are all valuable and very helpful for revising and improving our paper, as well as the important guiding significance to our researches. We have studied these comments carefully and have made corrections accordingly which we hope meet with your approval.

Reviewer #1 (Remarks to the Author):

Pseudomonas aeruginosa is an opportunistic bacterial pathogen that contains many virulence-associated genes. One of the genes is a phospholipase D enzyme called PldA that is more similar to eukaryotic PLDs than bacterial PLDs. Yang et al report the first high resolution crystal structure of *Pseudomonas aeruginosa* PldA that reveals a common PLD fold with a eukaryotic-like C-terminal domain, and a unique peripheral helical domain (PD domain) that appears to sit above the active site. A crystal structure of a truncated PldA is also determined. In this truncated structure, the catalytic domain is open to solvent with most of the flexible PD domain seemingly missing electron density, as well as a lid region in PldA. The authors next determine the structure of PldA bound to PA3488 using single particle cryoelectron microscopy. PA3488 binds adjacent to the catalytic domain and causes conformational changes in the lid region. PA3488 binds extremely tightly to PldA and mutations at the interface reduce the affinity of this interaction. Activity assays using a short chain lipid substrate show that in the presence of PA3488A the activity of PldA is reduced but not eliminated. Based on this structural work, a final model is proposed where PldA is initially bound to PA3488 and an unknown chaperone to inhibit PldA activity in a closed conformation, then secreted into a pre-open conformation, and finally adopting an open conformation that is a fully active PLD.

Overall the structure of this eukaryotic-like bacterial PLD is very interesting and there is a nice comparison with the eukaryotic PLDs in the discussion. However, some aspects are not clear and could be improved with modifications to the figures/text and additional experiments. For example, clarification of the biochemical assays and additional experiments would help support the proposed model, especially with regard to inhibition by PA3488 and the proposed increased activity in the open structure of PldA-truncated. Also, for conformational changes it is unclear if the conformational differences in the PldA-truncated structure are simply due to the lower resolution and lack of density for the PD region. Lastly, the exact roles of PldA and PA3488 in *P. aeruginosa* infection and virulence are unclear in some sentences.

Points to address

1. It is stated that PA3488 is an immunity protein of PldA that can neutralize

the virulence of PldA. However there is no reference for this statement, so it is unclear what evidence there is and how PA3488 impacts virulence.

Reply: We apologize for not making it clear in the original manuscript. In 2013, researches have proved PA3488 was sufficient to protect from PldA-dependent lysis by competitive lysis assays of *P. aeruginosa*¹. However, thus far, the detailed mechanism remains to be elusive, which is our primary purpose to explain the phenomenon at the molecule level. We have cited Alistair and colleagues' study in the revised version of the manuscript.

2. In the discussion, lines 289-290. It is stated that PldA is essential for virulence, but no reference is given. Previously (lines 88-92) it was stated that it has roles in inter-bacterial competition, eukaryotic host infection, and mammalian cell invasion. Does genetic deletion of PldA cause of loss of virulence? If not, the statement above should be toned back.

Reply: We thank the reviewer for pointing out the issue. Researchers have found that the virulence of *P. aeruginosa* dramatically decreased upon deletion of *pldA*¹, and previous studies also have proved that the virulence of *P. aeruginosa* is closely associated with PldA^{2,3}. We have rewritten it as “which is responsible for its virulence” in line 316 and added the references.

3. The authors describe the C-terminal domain as very similar to the eukaryotic human and plant PLD C-termini. However, this is not easily visualized in figure in 1c. Can this be more explicitly shown in a supplemental figure or new panel?

Reply: We apologize for not making it clear in the original manuscript. In the paper, we said the CCD of PldA, rather than the CTD, is very similar to its human and plant homologues (The RMSD of CCD is 10.753 between PldA and prokaryotic PLD, and less than 1 between PldA and eukaryotic PLDs). CCD contains the HKD1 and HKD2, their sequence alignment is shown in Extended Data Fig. 1a. To visualize the similarity of the CCD of PldA and its human and plant homologues more explicitly, we depicted a new panel in Extended Data Fig. 1b.

Extended Data Fig. 1|Conservation and stability of PldA. Structural comparisons of CCD of PldA with prokaryotic PLD (left panel, RMSD of 10.753 for 300 matched C α pairs) and eukaryotic PLDs (right panel, RMSD of 0.770 for 278 matched C α pairs of human and RMSD of 0.966 for 272 matched C α pairs of plant). *Streptomyces sp.*

(palecyan, PDB:1FOI), *Homo sapiens* (forest, PDB:6OHO) and *A. thaliana* (sand, PDB:6KZ9).

4. What regions of truncated PldA have been removed? Is loss of these regions the reason for the observed structural differences? Or are the structural differences due to the lower resolution and lack of electron density for the PD region?

Reply: We thank the reviewer for pointing out the issues. The crystal structure of PldA^{truncate} was determined using in situ proteolysis. The regions (aa 363-aa 411) have been removed in the PldA^{truncate}, which was updated in page 6 line 179-181. In the cryoEM map of PldA-PA3488 complex, there was no clear density to build the PD1 region (aa 239-aa 416). This region was resolved neither in the crystal structures nor in the cryoEM structure, which may be due to the flexibility in this region. As our knowledge, the missing of flexible regions don't affect the stable bulk of one structure in general.

5. The structure of truncated PldA has a calcium ion bound (table 1), and looking at the literature there is evidence that calcium increases the activity of PldA (see <https://doi.org/10.1046/j.1365-2958.2001.02282.x>). Does calcium increase the activity of PldA in the authors hands too? Was calcium included in the crystallization conditions? Where does this calcium bind and is the coordination consistent with calcium? How was this metal identified as calcium? Lastly, could the presence of calcium affect the conformation of PldA and its activity?

Reply: We thank the reviewer for valuable suggestions and questions. In the similar position in the plant PLD α ⁴, we found a metal ion which is absent in the crystallization reagents. Refer to the bond type and the structure of PLD α (which placed a Ca²⁺ here), we assigned it as a Ca²⁺. As shown in Extended Data Fig. 6a, comparing with PLD α , the Ca²⁺ in PldA is further away from the active site. It located between two surficial loops-L10 and L48 (An extended loop in PldA) and stabilized the region (Extended Data Fig. 6b). In addition, we have performed the enzymatic activity of PldA under the different concentrations of Ca²⁺, the results indicated Ca²⁺ doesn't affect the activity of PldA as shown in Extended Data Fig. 6c. It is probably because that the distance between the active site and the Ca²⁺ is too far. We also identified the existence of Ca²⁺ in PldA via (μ -XRF) analysis (Extended Data Fig. 6d). In the paper you mentioned⁵, researchers found that its activity was not significantly affected by low concentrated Ca²⁺ (between 0.1 mM and 1 mM, which is close to the normal physiological level) but was increased < 3.5-fold at high concentrated Ca²⁺ (5.0 mM, which is obviously higher than the normal physiological level), which is probably caused by other members of Ca²⁺-dependent phospholipase D. In *Arabidopsis*, scientists have found PLD zeta 1 is a calcium-independent phospholipase⁶. Likely, the activity of PldA doesn't depends on the concentration of Ca²⁺ as well. Here, we speculate Ca²⁺ may functions in stabling the crystal structure of PldA^{truncate}.

Extended Data Fig. 6 Ca^{2+} in the PldA structure. **a**, Comparison of the Ca^{2+} position in the PldA and PLD α . The results showed Ca^{2+} in the PldA is further away from the active site. **b**, The Ca^{2+} binding site in the PldA^{FL} structure. Four residues binding with Ca^{2+} and their distances are labeled. **c**, PldA enzyme activity measured with different concentrations of Ca^{2+} . Results are means \pm SD, $n = 3$. **d**, μ -XRF analysis of PldA. The purple peak in 3690 ev indicates the existence of Ca^{2+} .

6. The proposed model uses the open conformation of truncated PldA as a rationale for the proposed conformational changes. How does the activity of truncated PldA compare to the full-length protein? And is this consistent with the model?

Reply: Thank you for raising this important point. We have measured the activity of the truncated protein and found its activity is 2.7 times higher than that of PldA^{FL} (Extended Data Fig. 4), which is consistent with our conclusion in the manuscript.

Extended Data Fig. 4| In vitro kinetic assay of PldA and truncated PldA using di8:0-PC as substrate. a, Michaelis-Menten kinetics of PldA and PldA^{truncate}. b, The kinetic parameters were calculated and shown in the table.

6. It is stated that PA3488 is an inhibitor/chaperone of PldA, however PA3488 does not fully inhibit PldA (extended data Fig. 3C) with ~40% activity estimated to still remain. How do the authors rationalize this given the structural observations of the closed state.

Reply: Thank you for raising this important point. We have identified another protein-VgrG4b, which is encoded adjacent to PldA and can directly interact with it². It also inhibits the virulence of PldA. Further, when VgrG4b and PA3488 coexist, PldA's activity is almost lost as shown in Extended Data Fig. 3. So VgrG4b may be our speculated the "unknown chaperone" and the intermediate of PldA coupled with PA3488 and VgrG4b may be the "closed state" in our paper, which also meet physiological needs for blocking the virulence of PldA in *P. aeruginosa*. However, the detailed mechanism of PldA inhibited by VgrG4b needs to be investigated. That will be our next interested aspect.

Extended Data Fig. 3|Enzymatic assays of PldA. PA3488 and VgrG4bCt (aa 693-aa 808) completely inhibit the enzymatic activity of PldA.

7. Looking through the literature there was a previous, quite thorough, biochemical characterization of PldA activity by Spencer and Brown (DOI: 10.1021/bi501291t) using a liposome system, rather than short chain lipids. However, I do not see a reference to this work. From reading that paper, it seems PldA has a preference for phosphatidylethanolamine (PE) >>> phosphatidylcholine (PC), but can also hydrolyze all phospholipids except PI.

In this study, it is unclear how the activity assay was performed. The methods states that a short chain phosphatidic acid was used as a substrate (di8:0 PA). However, PA is the product, so I assume this must have been a typo. Was short chain PC used? Or another phospholipid? I assume PC since the amplex red assay was used to detect production of choline? How was "PC" solubilized? E.g. by BSA, liposomes, micelles, or just aqueous solvent since the acyl chains are short. More details should be included in the methods section and figure legends.

Given the prior literature, it seems most relevant to use PE as a substrate than PC. Another major concern is the use of a short chain lipid, especially given the lack of complete inhibition by PA3488. It seems more relevant to use a liposome system for activity assays to better reflect a true membrane environment and the effect of PA3488 on PldA activity and potentially membrane binding.

Reply: We thank the reviewer for pointing out this question. We indeed used di8:0 PC as substrate and PA was a typo. The detailed buffer dissolving PC was added in the text. And other details for enzyme activity assay and kinetic studies were also supplied.

We were sorry that we couldn't use a liposome system to test other substrate as we couldn't manage to get the relative materials in limited time. Therefore, the PldA activity was test according to another protocol using the Amplex Red Phospholipase D Assay Kit (Invitrogen, A12219). We have added the reference in the revised version.

8. In the cryoEM structure figure 4, it looks like there is a region modeled in the upper right panel (Fig. 4a) that does not have any cryo-EM density (right side, Fig. 4b). Is that correct? Or are the orientations different? It looks like this is region corresponds to the PD region.

Reply: We thank the reviewer for pointing out this question. Yes, the orientations are different. We have corrected the orientation of density maps and according models in the same views.

9. It would be useful to highlight where the active site of PldA is, in relation to bound PA3488.

Reply: We thank the reviewer for pointing out this and the suggestion is well taken. We have highlighted the active site of PldA in Fig. 4b in the revised version.

Fig. 4|Cryo-EM structure of PldA–PA3488 complex. b, Two orthogonal views of the model of the PldA–PA3488 complex are shown: PldA and PA3488 are colored in purple and green, respectively. The dotted regions indicate the active site.

10. It is stated that the full length PLD crystals had poor diffraction, so the truncated structure of PldA was determined. But the truncated structure is at a lower resolution. This seems opposite.

Reply: We thank the reviewer for pointing out this. Because of the large molecular weight and easy degradation of PldA, we worried that high quality crystal of PldA cannot be obtained. So we simultaneously screened the crystals of the full-length and the truncated PldA. To make it more explicit, we rewrote it as “PldA harbors a large molecular weight and degrades easily, considering the possible poor diffraction of PldA^{FL} crystals, we simultaneously determined the structure of PldA using in situ proteolysis (referred to as PldA^{truncate})” in the page 6, lines 176-178. Owing to good luck, the full-length PldA crystals surprisingly diffracted better than the truncated ones.

Reviewer #2 (Remarks to the Author):

Dear authors,

with great interest, I read your manuscript entitled "Structural insights into PA3488-mediated inactivation of *Pseudomonas aeruginosa* PldA". The manuscript presents two crystal structures of phospholipase D of *P. aeruginosa*, a full-length one and its truncated version, together with its structure bound to the PA3488 immunity protein, which was determined by cryo-electron microscopy. These three structures differ, highlighting the effects of the missing amino-acid chain and the PA3488 binding. Together with the knowledge of two phospholipid acid structures of *A. thaliana*, you proposed a model of how phospholipase D facilitates entry of *P. aeruginosa* pathogen into target cells.

In general, I believe that the presented research is solid and used appropriate

materials and methods to draw correct conclusions, which will be valuable for a broad range of readers and important in the fight with *Pseudomonas aeruginosa*. The manuscript is also written with good stylistics and has a very nice introduction section. However, in the current form it suffers from a large amount of ambiguities and requires thus a major revision.

Major points:

1. Starting model for model building of PldA: thank you for mentioning the difficulties with x-ray structure determination and molecular replacement. Did you try to confirm the resolved PldA structures with Alphafold PLdA structure of *P. aeruginosa* as a starting model?

Reply: We thank the reviewer for pointing out this. After Alphafold2 was released last year, we compared the current PldA structure in this manuscript with the PldA structure resolved by using the Alphafold2 predicted structure as a starting model, and found that the two structures are almost the same in general. Those regions, according with missing regions in our model, also have low credibility in the Alphafold2 predicted structure, which reflects these regions are very flexible and easy to degrade. To make it more explicitly, we depicted a new figure as shown in Extended Data Fig. 7 and added the explanation in the revised version lines 205-213.

Extended Data Fig. 7| AlphaFold2 predicted structure of PldA. a, Overall structure of AlphaFold2 predicted PldA in cartoon representation. Different colors indicate different pLDDT values and the two PD domains are marked. **b,** The PAE value of predicted PldA structure. **c,** Superposition of AlphaFold2 predicted PldA (green), PldA^{FL} (cyan) and PldA^{truncate}. **d-f,** Detailed comparison of different domains: The linker domain (**d**), PD1 domain (**e**) and PD2 domain (**f**).

2. CryoEM structure determination: What was your motivation to use data from all the last three iterations of local-search 3D classification and combine them later, discarding duplicates? Was it different from the results of the last iteration only? How many iterations were used in global and local runs? Please update the methods section accordingly. Which structure was used as initial 3D model? Could Bayesian polishing followed by another round of CTF refinement, or further 3D classification improve the map quality?

Reply: We thank the reviewer for pointing out these important questions. In our experience, to deal with the cryoEM data set of small complexes (<200 kDa), particles selection is very tricky because of lower SNR (signal noise ratio). Using the above strategy, it was different from the results of the last iteration only. Although most of good particles were still in good classes and most of bad particles remain in bad classes, some particles jumped between good and bad classes in different iteration.

We used 100 iterations in global angular searching 3D classification to make the initial angle information of each particle more accurate and more reliable. And then we used 25 iterations in local angular searching 3D classification to select good classes. We have updated the “Image processing” in the methods section. The initial model of the PldA-PA3488 complex was generated using 12,000 selected particles without reference in RELION 3.1 as described in the methods section.

We have tried Bayesian polishing, CTF refinement, and other 3D classification strategies. Also, CryoSPARC was used to process this data set. But the map quality was not improved, which may because this complex was only ~160 kDa affected the particle SNR.

3. It is totally unclear which amino-acids were removed from PldA to form the truncated version and with which motivation. In the current state, it is impossible to interpret the truncated structure and its differences to the full-length counterpart. Please issue this information upon the first mention of the truncated version.

Reply: We thank the reviewer for pointing out this important point. In the PldA^{truncate}, the region including aa 363- aa 411, are removed and was updated in page 6 line 179-181. Because of the large molecular weight and easy degradation of PldA, we worried that high quality crystal of PldA cannot be obtained. So we simultaneously screened the crystals of the full-length and the truncated PldA. We rewrote it as “PldA harbors a large molecular weight and degrades easily, considering the possible poor diffraction of PldA^{FL} crystals, we simultaneously determined the structure of PldA using in situ proteolysis (referred to as PldA^{truncate})” in the page 6, lines 176-178. In addition, we

have measured the enzymatic activity of the PldA^{truncate} and found it has a higher activity than PldA^{FL} as shown in Extended Data Fig. 4, which is consistent with our conclusion. We have added above information in the revised version.

Extended Data Fig. 4| In vitro kinetic assay of PldA and truncated PldA using di8:0-PC as substrate. a, Michaelis-Menten kinetics of PldA and PldA^{truncate}. b, The kinetic parameters were calculated and shown in the table.

4. Inspecting the submitted pdb files, I noted a couple of unmodeled regions in all structures. Please, disclose this essential information. In particular: why are these residues missing in the cryoEM model and the full-length model? In particular, could AA 239-416 in cryoem structure, AA 396-416 in full-length structure could play a role in the structure? Please comment.

Reply: We thank the reviewer for pointing out this. Due to the lack of density in maps resolved by crystallography and electron microscopy, the couple of unmodeled regions in all structures can't be built, which suggests these regions are very flexible and unstable. Thus far, what is the function of the missing region is not clear. It also is an interesting question in our future plan.

5. I am missing a figure showing the model fitted to the cryoEM density in the region of the active residues of the catalytic site and the PldA-PA3488 interface. Please add such figure.

Reply: We thank the reviewer for pointing out this and the suggestion is well taken. We have highlighted the active site of PldA in Fig. 4b in the revised version to make the active site of PldA and the interface of PldA-PA3488 more explicit in the revised version.

Fig. 4|Cryo-EM structure of PldA–PA3488 complex. b, Two orthogonal views of the model of the PldA–PA3488 complex are shown: PldA and PA3488 are colored in purple and green, respectively. The dotted circle regions indicate the active site.

6. In general, the manuscript suffers from the lack of correct referencing and unique and consequent shorning nomenclature. For a non-expert reader, it is difficult to discriminate among PldA, PID alpha, hPLD, etc. Please provide standard species naming upon first mention in the manuscript.

Reply: We thank the reviewer for pointing out this and the suggestion is well taken. We have provided standard species naming in the revised version upon first mention in the manuscript.

7. Together with point 6, it is essential to provide PDB codes or references for all structures mentioned. Which structure was used to model the PA3488 protein?

Reply: We thank the reviewer for pointing out this and the suggestion is well taken. We have provided PDB codes or references for all structure mentioned in the revised version.

8. "PA8" misses a reference. I assume it is phosphatidic acid (PA stands already in the introduction, indeed). What is the "8"? Could you please compare the position of the docked PA in *A. thaliana* structure with your full-length and cryoem structures?

Reply: We thank the reviewer for pointing out these questions and the suggestions is well taken. The number "8" in the "PA8" represents the length of fatty acyl chain in phosphatidic acid. We have provided the standard name upon first mention in the revised version. We depicted a new panel to show the position of PA8 in *A. thaliana* PLD-PLD α (6KZ8) in Fig. 2d. In addition, we also attempted to dock PA8 into PldA inhibited by PA3488, but the experiment failed due to the PA3488-inhibited PldA was in a more closed state. So we can't show the position of PA8 in PA3488-inhibited PldA.

Fig. 2| Catalytic pocket of PldA. **a**, The hydrogen-bonding network present in the active site of PldA. Several water molecules (W1-W6) are involved in the hydrogen bond interaction with active site of PldA. **b**, Comparison of the lid regions of PldA^{FL} (slate) with closed (sand) and open PLD α (dark teal). **c-f**, The docking result shows the binding of PA8 in the catalytic pocket of PldA^{FL} (c), and surface views showing the binding of PA8 in the catalytic pocket of PLD α (6KZ8) (d), PldA^{FL} (e) and PldA^{truncate} (f).

9. I find the structural differences between *A. thaliana* and *P. aeruginosa* "lid" regions rather large, and would hesitate to join them into one common mechanism. I suggest to refer to each "closed" or open" structure separately rather than to "pre-open" and "pre-closed", and compare the two species PldAs

afterwards. Could you please also comment on sequence similarity in the lid region?

Reply: We thank the reviewer for pointing out this. In this paper, we clearly found the lid regions of PldA^{FL} and PldA-PA3488 are located between the “open” and “closed” states reported in *A. thaliana*. The state of PldA^{FL} is closely to the “open”, and that of PldA-PA3488 is closely to the “closed”, hereafter as referred “pre-open” (Fig. 2b) and “pre-closed” (Fig. 5c). The lid region of PldA^{truncate} maybe the “open” state and subsequent enzymatic activity assay proved our speculation as shown in Extended Data Fig. 4. In addition, PA3488 don’t completely block the activity (Extended Data Fig. 3c). Thus, we speculate there is another chaperone, together with PA3488, enable PldA the “closed” state. Indeed, we found the activity of PldA is almost lost upon PA3488 and VgrG coexist as shown in Extended Data Fig. 3. So it is rational for the understanding of mechanism. We also performed the sequence alignment of lid region as shown in Extended Data Fig. 2b. The results suggest the lid region has low sequence similarity among different species, which may reflect its species-specificity as the different responding mechanism for Ca²⁺ concentration. We have added the above information in the revised version.

Extended Data Fig. 4| In vitro kinetic assay of PldA and truncated PldA using di8:0-PC as substrate. a, Michaelis-Menten kinetics of PldA and PldAtruncate. **b,** The kinetic parameters were calculated and shown in the table. The experiments were repeated three times with similar results.

Extended Data Fig. 3| Enzymatic assays of PldA. PA3488 and VgrG completely inhibit the enzymatic activity of PldA.

PldA(117-130)	- - - - -	S	P	S	G	S	L	G	T	Y	D	F	E	T	M			
PLD1(422-435)	- - -	E	V	E	L	A	L	G	I	N	S	E	Y	T	K	- - -		
PLD2(401-417)	L	F	K	E	V	E	L	A	L	G	I	N	S	G	Y	S	K	- - -
PLD α (283-295)	- - - - -	K	K	D	G	L	M	A	T	H	D	E	E	T	-			
bPLD(125-139)	- -	A	P	V	Y	H	M	N	G	I	P	S	K	Y	R	D	- - -	

Extended Data Fig. 2| Comparison of active pockets from PLDs. Sequence alignment of lid region from different species.

Minor points:

1. Figure legends are in general very brief, please update them with PDB codes to the structures shown.

Reply: We thank the reviewer for pointing out this and the suggestion is well taken.

2. Legend text to Ext. Figure 3C is missing.

Reply: We thank the reviewer for pointing out this. We have updated the legend text to the Extended Data Fig. 3c.

3. Figures also typically lack descriptors and amino acid coding, which makes it impossible to understand them. On top of it, I would suggest to switch to ribbon representation if structural differences are depicted, or compare the structures side by side (e.g. Fig. 1C, 2B, 5A). Please, consider able to enlarge the field of view a bit so that the reader can get more context information.

Reply: We thank the reviewer for pointing out these questions and the suggestions are well taken. For Fig. 2b and Fig. 5a, we have added the descriptors and amino acid coding and switch to ribbon representation. For Fig. 1c, we have depicted a new panel in Extended Data Fig. 1f-h to make them more clearly for readers.

Extended Data Fig. 1| Conservation and stability of PldA. f-h, Structures of PLDs from different species including PldA (f), hPLD2 (g) and PLD α (h).

4. Introduction, p. 2, l. 58: please name some of the diseases

Reply: We thank the reviewer for pointing out this and the suggestion is well taken.

5. Results, p. 4., l 107-110: please include the AA ranges for each domain, and depict the core catalytic domain in Fig. 1A.

Reply: We thank the reviewer for pointing out this and the suggestion is well taken.

6. Results, p. 4., l. 120: Which B-factor is meant?

Reply: We thank the reviewer for pointing out this. B factor is a standard parameter to represent the degree of dispersion of an atom, and it reflects the stability of structure. The score of B factor is negatively correlated with the stability. We have explained it in the revised version.

7. Results, Eukaryotic-like catalytic core of PldA: It would be nice to state that both HKDs co-form to it right at the beginning.

Reply: We thank the reviewer for pointing out this and the suggestion is well taken.

8. results, p. 5, l. 156: add reference to the corresponding figure.

Reply: We thank the reviewer for pointing out this and the suggestion is well taken.

9. Results, Substrate binding pocket: Please indicate AA involved.

Reply: We thank the reviewer for pointing out this and the suggestion is well taken.

10. Results, p. 5, l. 160: bPLD - please indicate species.

Reply: We thank the reviewer for pointing out this and the suggestion is well taken.

11. Results, p. 7, l. 210-211: mention AA involved in the lid. Fig. 4a doesn't show it.

Reply: We thank the reviewer for pointing out this and the suggestion is well taken.

12. Discussion, p. 10, l. 319-321: please include reference articles.

Reply: We thank the reviewer for pointing out this and the suggestion is well taken.

13. Fig. 2: describe water molecules and bond types in the legend.

Reply: We thank the reviewer for pointing out this and the suggestion is well taken.

14. Ex. Fig.1 b: color code missing. d: indicate position of PD helices in the figure.

Reply: We thank the reviewer for pointing out this and the suggestion is well taken.

With best regards!

Reviewer #3 (Remarks to the Author):

The authors describe an interesting study regarding the *P. aeruginosa* phospholipase D, PldA. The combined crystallographic/cryo-EM analysis of PldA with and without its immunity protein PA3488 yielded insights into the activation and inhibition of this protein. Although the study is quite interesting and the structural biology aspects of this study are adequately performed, there are many other serious issues that need to be addressed and a major revision is therefore requested.

Major shortcomings:

-Calcium is known to activate some bacterial phospholipase Ds, briefly mentioned on line 83. (but also PMC3233269); the authors did not probe this in their assays and should have performed such a study. Also, there is one Ca²⁺ in the 2nd crystal structure (truncated) yet the position of this calcium is not described. What is the role of Ca²⁺ in PLD activity? How does the activity of PLD depends on the concentration of Ca²⁺, where none of these catalytic residues (HKD) show binding of Ca²⁺? Does Ca²⁺ support structural stabilization of PLD?

Reply: We thank the reviewer for valuable suggestions and questions. We have performed the enzymatic activity of PldA under the different concentrations of Ca²⁺, the results indicated Ca²⁺ doesn't affect the activity of PldA as shown in Extended Data Fig. 6. So the activity of PldA doesn't significantly depends on the concentration of Ca²⁺. As what you said, we speculate Ca²⁺ may functions in stabling the structure of PldA.

Extended Data Fig. 6| PldA enzyme activity measured with different concentrations of Ca²⁺. Results are means \pm SD, n = 3.

-the catalytic activity of the truncated protein (obtained by limited proteolysis) was never measured and is needed for this structure-function study.

Reply: We thank the reviewer for pointing out the important question. We have

measured the activity of the truncated protein and found its activity is higher than the full-length of PldA as shown in Extended Data Fig. 4, which is consistent with our conclusion in the manuscript.

Extended Data Fig. 4| In vitro kinetic assay of PldA and truncated PldA using di8:0-PC as substrate. a, Michaelis-Menten kinetics of PldA and PldAtruncate. **b,** The kinetic parameters were calculated and shown in the table.

-the authors bring up that there is an unknown chaperone but do not provide any evidence (from the literature and/or is it in their cryo-EM structure?).

Reply: Thank you for raising this important point. We have identified another protein-VgrG4b (PA3486), which is located on the injection apparatus of type VI secret system in the intracellular of *P. aeruginosa*. It also inhibits the virulence of PldA. Further, when VgrG4bCt (aa 693-808) and PA3488 coexist, PldA's activity is almost lost as shown in Extended Data Fig. 3. So VgrG4b may be our speculated "unknown chaperone" and the intermediate of PldA coupled with PA3488 and VgrG4b may be the "closed state" in our paper. We have added the associated information in the revised version. However, the detailed mechanism of PldA inhibited by VgrG4b need to investigate. That will be our next interested aspect.

Extended Data Fig. 3| Enzymatic assays of PldA. PA3488 and VgrG4bCt completely inhibit the enzymatic activity of PldA.

-the raw SPR data for the curves is not presented (only final K_a and K_d values are listed).

Reply: We thank the reviewer for pointing this. The raw SPR data for curves are

supplied in Extended data Fig. 9

-there need to be replicates in both the SPR measurements and activity measurements to improve scientific rigor (p-values, standard deviations).

Reply: We thank the reviewer for pointing this, the activity measurements are all updated with error bars indicating mean SD with three replicates.

-In the discussion section, the authors compare the different PLDs with some domains being present and others absent. Although surface representations of these proteins are included in Extended data Fig. 2, it would be helpful if also cartoon representations of these proteins are shown with all the different domains colored consistently and the orientations are the same so the reader can follow the discussion.

Reply: We thank the reviewer for pointing this. In Extended data Fig. 2, we mainly told readers the information about the changes of lid regions and each pocket in PLDs. The conformational changes of domains in different PLDs have been showed in Figs. 2-5. We tried to color different domains by different colors, however, by doing that, the pockets are not easy to distinguish. Besides, the color scheme in this figure is consistent with other figures. So we retained the current fashion.

-it would be informative if a Mass Spectrometry analysis were done on the crystals of the truncated protein to figure out the precise starting and ending amino acid.

Reply: We thank the reviewer for pointing this. After using a time of flight mass spectrometer (MALDI-TOF/TOF Ultraflextreme™, Bruker, Germany), we figured out the precise truncated region of PldA and updated in line 179-181.

-Did the authors check the crystal contacts, while comparing the movement of the α 11-helix as well other conformational changes to rule out the possibility of crystallization artifact of PldA(FL) and PldA(truncated) proteins in two different space groups?

Reply: Thanks for the suggestion concerning crystal contacts raised by truncation and different space groups. By inspection of the crystal contacts, the α 11 does play a role in stabilizing the conformation of truncated PldA through neighboring α 18' and α 19' under the captured unit cell as shown in Extended Data Fig. 5. Considering that the truncated PldA achieved higher catalytic activity (Extended Data Fig. 4) than the full-length PldA and the α 11 is relatively flexible anchoring on a linker (residues Thr498 and Pro499), and combining with three states resolved in this paper and the alphafold2 predicted structure (Extended Data Fig. 7), it is almost confirmed that the α 11 could flip flexibly. Scrupulously, we emphasized that the 120°-flipped α 11 is stabilized by crystal contacts in the main text.

Extended Data Fig. 4| In vitro kinetic assay of PldA and truncated PldA using di8:0-PC as substrate. a, Michaelis-Menten kinetics of PldA and PldA^{truncate}. b, The kinetic parameters were calculated and shown in the table. The experiments were repeated three times with similar results.

Extended Data Fig. 5| $\alpha 11$ stabilizes the structure of PldA^{truncate}. a, The flipped $\alpha 11$ is stabilized by the $\alpha 18'$ and $\alpha 19'$ form a symmetric PldA' in the unit cell. b, $\alpha 11$ anchors on the Thr498 and Pro499 to make it flip flexibly.

Extended Data Fig. 7 | AlphaFold2 prediction structure of PldA. **a**, Overall structure of AlphaFold2 predicted PldA in cartoon representation. Different colors indicate different pLDDT values and the two PD domains are marked. **b**, The PAE value of predicted PldA structure. **c**, Superposition of AlphaFold2 predicted PldA (green), PldAFL(cyan) and PldA^{truncate}. **d-f**, Detailed comparison of different domains: The linker domain (**d**), PD1 domain (**e**) and PD2 domain (**f**).

-How the molecular surfaces and substrate-binding pockets of PldA, hPLD2, PLD α , and bPLD were compared? No surface cavity calculations were performed.

Reply: Thank you for raising this important point. We have calculated surface cavity of different PLDs through a web server- CASTp 3.0⁷ and added the volume of each pocket in Extended data Fig. 2.

Extended Data Fig. 2 | Comparison of active pockets from PLDs. **a**, Alignment of conserved active sites in the HKD domain from PLDs (PldA, hPLD2, and PLDα are colored as in Fig. 1C, bPLD is colored pale cyan). **b**, Sequence alignment of lid region from different species. **c-g**, Surface (upper) and cross-section (bottom) representation of active pockets from PLDs. The volume of each pocket in PLDs is calculated through a web server-CASTp 3.0. The significant conformational change of the lid region is highlighted in yellow and the active site is highlighted in red.

-The authors mention, the binding of PA3488 inactivates PldA, however in "Extended Data Fig. 3", panel C, the PldA+PA3488 shows partial activity. The authors should provide an explanation as the affinity between the proteins is

less than 0.1 nM.

Reply: We thank the reviewer for pointing out the important question. As what you said, in this study, PA3488 can't fully neutralize the enzymatic activity of PldA, So we speculated there is another "unknown chaperone" can further decrease the activity of PldA. Indeed, we found VgrG4b (PA3486) also inhibited partial activity of PldA as shown in Extended Data Fig. 3. Unexpectedly, the activity of PldA is completely lost upon VgrG4bCt and PA3488 coexist, which is consistent with our speculation. In addition, the high affinity between PldA and PA3488 suggests its activity needs to be inhibited by coupled PA3488 and/or VgrG4b before PldA is secreted out the cell.

Extended Data Fig. 3| Enzymatic assays of PldA. PA3488 and VgrG completely inhibit the enzymatic activity of PldA.

Minor comments:

-line 46: replace "helices" with "helical"

Reply: We thank the reviewer for pointing out this and the suggestion is well taken.

-line 46: replace "mutation assay" with "mutational analysis"

Reply: We thank the reviewer for pointing out this and the suggestion is well taken.

-line 46: replace "Otherwise" with "In contrast"

Reply: We thank the reviewer for pointing out this and the suggestion is well taken.

-line 77: "two HKD motifs located in the C-terminal catalytic domain". HKD motifs located well in the middle of the sequence.

Reply: We thank the reviewer for pointing out this and the suggestion is well taken.

-line 85: it is not clear what the authors mean. Perhaps it is better to replace "low" with "some detectable" if that is what the authors mean to convey (i.e., PldA does not have any detectable sequence identity? Or do the authors mean that PldA has high sequence identity).

Reply: We apologize for not making it clear in the original manuscript. Here, we mean prokaryotic PLDs normally have low sequence identity to the eukaryotic PLDs, but PldA has higher sequence identity to the eukaryotic PLDs compared to the prokaryotic

PLDs. We have rewritten it in the revised version to make it clearly.

-line 114: replace "almost same as" with "very similar to"

Reply: We thank the reviewer for pointing out this and the suggestion is well taken.

-line 119: replace "periphery" with "peripheral"

Reply: We thank the reviewer for pointing out this and the suggestion is well taken.

-line 122: replace "these" with "the positions of these". Also, delete the word "integrally"

Reply: We thank the reviewer for pointing out this and the suggestion is well taken.

-line 135: add "of wt activity" after "30%"

Reply: We thank the reviewer for pointing out this and the suggestion is well taken.

-line 137: add the word "region" after "C-terminal"

Reply: We thank the reviewer for pointing out this and the suggestion is well taken.

-line 143: hydroxy should be hydroxyl

Reply: We thank the reviewer for pointing out this and the suggestion is well taken.

-line 145: "binding" should be "bonding"

Reply: We thank the reviewer for pointing out this and the suggestion is well taken.

-line 150: replace "well" with "site"

Reply: We thank the reviewer for pointing out this and the suggestion is well taken.

-line 152: add "site" after "active"

Reply: We thank the reviewer for pointing out this and the suggestion is well taken.

-line 153: "configuration" should be "conformation"

Reply: We thank the reviewer for pointing out this and the suggestion is well taken.

-line 170: replace "against" with "at"

Reply: We thank the reviewer for pointing out this and the suggestion is well taken.

-line 192: PA8 is never defined, add a definition as people outside this field do not know what that is.

Reply: We thank the reviewer for pointing out this and the suggestion is well taken.

-line 198: replace "near" with "nearby"

Reply: We thank the reviewer for pointing out this and the suggestion is well taken.

-line 222: replace "stretches" with a different word

Reply: We thank the reviewer for pointing out this. We have replaced "stretches" with "concave surface".

-line 338: replace "dominant" with "large"

Reply: We thank the reviewer for pointing out this and the suggestion is well taken.

-line 661: PldAD643+AR820+AR969A. Please correct the mutants R820A and R969A

Reply: We thank the reviewer for pointing out this. We have corrected them in the revise version.

-Extended Data Fig. 3- Description of 'c' panel missing.

Reply: We thank the reviewer for pointing out this. We have updated the legend text to the Extend Figure 3C.

References

- 1 Russell, A. B. *et al.* Diverse type VI secretion phospholipases are functionally plastic antibacterial effectors. *Nature* **496**, 508-512, doi:10.1038/nature12074 (2013).
- 2 Jiang, F., Waterfield, N. R., Yang, J., Yang, G. & Jin, Q. A *Pseudomonas aeruginosa* type VI secretion phospholipase D effector targets both prokaryotic and eukaryotic cells. *Cell Host Microbe* **15**, 600-610, doi:10.1016/j.chom.2014.04.010 (2014).
- 3 Boulant, T. *et al.* Higher Prevalence of PldA, a *Pseudomonas aeruginosa* Trans-Kingdom H2-Type VI Secretion System Effector, in Clinical Isolates Responsible for Acute Infections and in Multidrug Resistant Strains. *Front Microbiol* **9**, 2578, doi:10.3389/fmicb.2018.02578 (2018).
- 4 Li, J. *et al.* Crystal structure of plant PLD α 1 reveals catalytic and regulatory mechanisms of eukaryotic phospholipase D. *Cell Res* **30**, 61-69, doi:10.1038/s41422-019-0244-6 (2020).
- 5 Wilderman, P. J., Vasil, A. I., Johnson, Z. & Vasil, M. L. Genetic and biochemical analyses of a eukaryotic-like phospholipase D of *Pseudomonas aeruginosa* suggest horizontal acquisition and a role for persistence in a chronic pulmonary infection model. *Mol Microbiol* **39**, 291-303, doi:10.1046/j.1365-2958.2001.02282.x (2001).
- 6 Qin, C. & Wang, X. The Arabidopsis phospholipase D family. Characterization of a calcium-independent and phosphatidylcholine-selective PLD zeta 1 with distinct regulatory domains. *Plant Physiol* **128**, 1057-1068, doi:10.1104/pp.010928 (2002).
- 7 Tian, W., Chen, C., Lei, X., Zhao, J. & Liang, J. CASTp 3.0: computed atlas of surface topography of proteins. *Nucleic Acids Res* **46**, W363-W367, doi:10.1093/nar/gky473 (2018).

Reviewer #1 (Remarks to the Author):

The authors have addressed all my initial comments appropriately.

The authors may consider starting a new paragraph on line 95, that would begin with the sentence "Here ..."

There appears to be a minor typo in line 137, activity.

Reviewer #2 (Remarks to the Author):

Dear authors,

many thanks for all your efforts that you have invested into the manuscript revision. I find you have addressed all my points I had raised correctly, the text has greatly improved and your story has become much more easier to understand now, making it a very pleasing reading. Nonetheless, I have still found a few mistakes and have one more topic to consider.

Major point:

It seems that there are two allosteric mechanisms that provide regulation to the active site: via the C-termini residues 1070-1073, and via the lid region formed by the N-termini residues 117-130, which change conformation upon binding of the PA3488 protein, and become flexible in the truncated construct. Both mutations of the C-termini and the PA3488 binding seem to reduce the enzymatic activity of PLDA by roughly the same amount (Fig. 1e, Ext. Fig 3c).

1) I think that Ext. Fig. 3c is an important result and should be presented in figure 4 or 5.

2) Similarly to mutating amino acids in the N-terminal tail, point mutations of the 3 amino acids 120-122 should modify/restore the enzyme activity. Have you performed such experiment? It may yield an interesting supplemental information to your story.

3) Checking with Matchmaker in UCSF Chimera, it seems that it is the Leu122 that rotates into the entrance of the active site, rather than that the whole lid region moves closer to it. If I am right, can you please rectify it in p. 9, l. 278-280? Together with this, I think it would be helpful to merge Figure 5b with Ext. Fig. 2c-d, and show the surfaces together in Fig.5.

4) Unfortunately, none of your figures clearly shows the spatial organisation of the C-terminus (AA 1070-1073) and the lid region with respect to the active site. Could you please indicate the positions of all three by color code in Fig. 1 b?

Minor points:

1) p. 4, l. 105: Arabidopsis -> A. thaliana

2) p. 5, l. 156: comparison with -> comparison to

3) p. 5, l. 159: PLDA α -> A. thaliana PLDA α , PLDA -> PLDA^{FL}

l. 160: active center -> active center of PLDA α

l. 165: Arabidopsis thaliana -> Streptomyces. Which species exactly?

4) truncated PLDA, p. 6., l. 176-196: The readers would deserve information about PLDA AA regions, which couldn't be modeled due to flexibility. In particular, the lid region and AA 221-451 of PD1.

l. 192: alpha11 helix -> alpha11 helix of truncated PLDA

5) p. 8, l. 237. Fig. 7 -> Fig.5

6) Fig. 2 legend: PLDA α -> A. thaliana PLDA α

7) p. 25, l. 641: Table 1 -> Extended Table 1
l. 679: CTF correction -> CTF estimation
p. 26, l. 689: For the last three -> For each of the last three

8. Ext. Fig. 6: Add a label for the Ca atom. Caption: Ca²⁺ in the TRUNCATED PLDA structure

With best regards!

Reviewer #3 (Remarks to the Author):

The authors adequately addressed the comments of the reviewers and the manuscript can now be accepted.

Thank you for your letter and for the reviews' nice comments on our article. According to your suggestions, we have corrected several mistakes and supplemented several data in our previous draft. Based on the comments, we also attached a point-by-point responses to the other two reviews as the following:

Reviewer #1 (Remarks to the Author):

The authors have addressed all my initial comments appropriately.

The authors may consider starting a new paragraph on line 95, that would begin with the sentence "Here ..."

Reply: We thank the reviewer for pointing out this and the suggestion is well taken.

There appears to be a minor typo in line 137, activity.

Reply: We thank the reviewer for pointing out this and the typo has been corrected in the revised version.

Reviewer #2 (Remarks to the Author):

Dear authors,

many thanks for all your efforts that you have invested into the manuscript revision. I find you have addressed all my points I had raised correctly, the text has greatly improved and your story has become much more easier to understand now, making it a very pleasing reading. Nonetheless, I have still found a few mistakes and have one more topic to consider.

Major point:

It seems that there are two allosteric mechanisms that provide regulation to the active site: via the C-termini residues 1070-1073, and via the lid region formed by the N-termini residues 117-130, which change conformation upon binding of the PA3488 protein, and become flexible in the truncated construct. Both mutations of the C-termini and the PA3488 binding seem to reduce the enzymatic activity of PLDA by roughly the same amount (Fig. 1e, Ext. Fig 3c).

1) I think that Ext. Fig. 3c is an important result and should be presented in figure 4 or 5.

Reply: We thank the reviewer for pointing out this and the suggestion is well taken. We have combined the results of Extended Fig. 3c into the Fig. 5 in the revised version.

Fig. 5 | PA3488 binding results in conformational changes in the structure of PldA and associated enzymatic assays. f, Structural comparison of apo-PA3488 (PDB: 5XMG) and PldA-bound PA3488. **g**, Inhibition of PldA activity by PA3488 and VgrG4bCt (aa 693-808). Results are means \pm SD, n = 3.

2) Similarly to mutating amino acids in the N-terminal tail, point mutations of the 3 amino acids 120-122 should modify/restore the enzyme activity. Have you performed such experiment? It may yield an interesting supplemental information to your story.

Reply: We thank the reviewer for the suggestion. Sorry, we haven't performed the point mutations of the aa120-122 yet. Your suggestion will be considered carefully in our next plan.

3) Checking with Matchmaker in UCSF Chimera, it seems that it is the Leu122 that rotates into the entrance of the active site, rather than that the whole lid region moves closer to it. If I am right, can you please rectify it in p. 9, l. 278-280? Together with this, I think it would be helpful to merge Figure 5b with Ext. Fig. 2c-d, and show the surfaces together in Fig.5.

Reply: We thank the reviewer for pointing out these questions. We have checked Leu122 in the lid carefully once again and it is convinced that the whole lid region (aa120-122) gradually rotates into the entrance of the active site from the open state, the pre-open state to the pre-closed state. In addition, we have merged the Extended Fig. 2c-d into Figure 5b to show their surface together.

Fig. 5|PA3488 binding results in conformational changes in the structure of PldA and associated enzymatic assays. b, Surface (up) and cross-sectional (down) representation of active pockets from PldA^{truncate}, PldA^{FL} and PA3488-inhibited PldA. The lid region and the active site are highlighted in yellow and red respectively. The volume of each pocket in PLDs is calculated through a web server-CASTp 3.0.

4) Unfortunately, none of your figures clearly shows the spatial organisation of the C-terminus (AA 1070-1073) and the lid region with respect to the active site. Could you please indicate the positions of all three by color code in Fig. 1 b?

Reply: We thank the reviewer for pointing out this. We have indicated these positions by color code in Figure 1b in the revised version.

Fig. 1|Overall structure of PldA^{FL}. b, Model of PldA with two views. The colors of the domains are the same as those in a.

Minor points:

1) p. 4, l. 105: Arabidopsis -> A. thaliana

Reply: We thank the reviewer for pointing out this and the suggestion is well taken.

2) p. 5, l. 156: comparison with -> comparison to
Reply: We thank the reviewer for pointing out this and the suggestion is well taken.

3) p. 5, l. 159: PLDalpha -> A. thalina PLDalpha, PLDA -> PLDA^{FL}
l. 160: active center -> active center of PLDalpha
l. 165: Arabidopsis thaliana -> Streptomyces. Which species exactly?
Reply: We thank the reviewer for pointing out these and the suggestions are well taken.

4) truncated PLDA, p. 6., l. 176-196: The readers would deserve information about PLDA AA regions, which couldn't be modeled due to flexibility. In particular, the lid region and AA 221-451 of PD1.

l. 192: aplha11 helix -> alpha11 helix of truncated PLDA
Reply: We thank the reviewer for pointing out these and the suggestions are well taken.

5) p. 8, l. 237. Fig. 7 -> Fig.5
Reply: We thank the reviewer for pointing out this and the suggestion is well taken.

6) Fig. 2 legend: PLDalpha -> A. thaliana PLDalpha
Reply: We thank the reviewer for pointing out this and the suggestion is well taken.

7) p. 25, l. 641: Table 1 -> Extended Table 1
l. 679: CTF correction -> CTF estimation
p. 26, l. 689: For the last three -> For each of the last three
Reply: We thank the reviewer for pointing out these and the suggestions are well taken.

8. Ext. Fig. 6: Add a label for the Ca atom. Caption: Ca²⁺ in the TRUNCATED PLDA structure
Reply: We thank the reviewer for pointing out these and the suggestions are well taken.

With best regards!